# UniBench: Visual Reasoning Requires Rethinking Vision-Language Beyond Scaling

**Haider Al-Tahan**[1], **Quentin Garrido**[1,2], **Randall Balestriero**[3],
**Diane Bouchacourt**[1], **Caner Hazirbas**[1], **Mark Ibrahim**[1]
[1]Meta FAIR, [2]Univ Gustave Eiffel, CNRS, LIGM, [3]Brown University

## Abstract

Significant research efforts have been made to scale and improve vision-language model (VLM) training approaches. Yet, with an ever-growing number of benchmarks, researchers are tasked with the heavy burden of implementing each protocol, bearing a non-trivial computational cost, and making sense of how all these benchmarks translate into meaningful axes of progress. To facilitate a systematic evaluation of VLM progress, we introduce `UniBench`: a unified implementation of 50+ VLM benchmarks spanning a range of carefully categorized vision-centric capabilities from object recognition to spatial awareness, counting, and much more. We showcase the utility of `UniBench` for measuring progress by evaluating nearly 60 publicly available vision-language models, trained on scales of up to 12.8B samples. We find that while scaling training data or model size can boost many vision-language model capabilities, scaling offers little benefit for reasoning or relations. Surprisingly, we also discover today's best VLMs struggle on simple digit recognition and counting tasks, e.g. MNIST, which much simpler networks can solve. Where scale falls short, we find that more precise interventions, such as data quality or tailored-learning objectives offer more promise. For practitioners, we also offer guidance on selecting a suitable VLM for a given application. Finally, we release an easy-to-run `UniBench` code-base with the full set of 50+ benchmarks and comparisons across 59 models as well as a distilled, representative set of benchmarks that runs in 5 minutes on a single GPU. *UniBench* with model evaluations on all benchmarks are provided as a toolbox at: https://github.com/facebookresearch/unibench

## 1   Introduction

The growing investment in vision-language models (VLMs), capable of a range of open-world multimodal tasks, has spurred the development of numerous benchmarks. Although in principle a more thorough set of evaluations is welcome, the ever-growing number of benchmarks has resulted in a complex, fragmented landscape for evaluation. Researchers are tasked with the heavy burden of implementing the protocol for each benchmark and making sense of how all these benchmarks translate into meaningful axes of progress. Of course, running such a large number of benchmarks also carries a non-trivial computational burden. Consequently, many new models are evaluated only on a *subset of available benchmarks*. When benchmarks are omitted, the research community is faced with blind spots in model strengths and weaknesses. Additionally, comparing the performance of one model versus others becomes challenging as the underlying set of benchmarks is not comparable. Ultimately, drawing well-founded conclusions about the best strategies to advance VLMs in this fragmented landscape of benchmarks is a challenge.

To help researchers navigate this overwhelming landscape of benchmarks and ease the burden of systematically evaluating VLMs, we introduce `UniBench`. In `UniBench` we implement 53 vision-

38th Conference on Neural Information Processing Systems (NeurIPS 2024) Track on Datasets and Benchmarks.

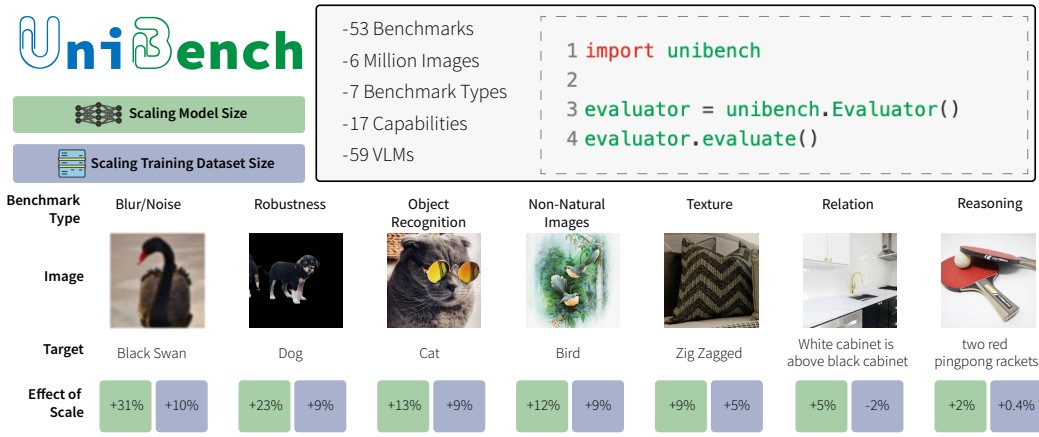

Figure 1: **Benchmark Types in `UniBench` with their respective performance gains from scaling model size and training dataset size.** Scale offers limited benefits for relational understanding and reasoning tasks.

language model benchmarks in a unified, user-friendly code-base. These benchmarks cover a range of vision-centric capabilities from standard object recognition to spatial understanding, counting, geographic robustness, domain-specific medical and satellite imagery, and many others. With such a comprehensive set of benchmarks, we shine a light on the blind spots in the strengths and weaknesses of the model. Next, to ensure that the research community can translate the many resulting metrics into meaningful axes of progress, we categorize these benchmarks into seven types and seventeen finer-grained capabilities, as shown in Figure 1. Researchers can quickly pinpoint model strengths and weaknesses in a comprehensive, apples-to-apples fashion.

We demonstrate the utility of `UniBench` by evaluating nearly 60 openly available vision-language models spanning a range of architectures, model sizes, training dataset scales, and learning objectives with scales of up to 12.8B samples and 1B parameters. We systematically compare this diverse set of models across the axes of progress in `UniBench`. We find that scaling, model size, or training data is a powerful lever for many axes of performance, but offers little benefit for visual relations and reasoning. We also find today's best VLMs struggle with simple benchmarks involving numerical comprehension, even with the right training data, on tasks such as character recognition or counting—including decades old benchmarks such as MNIST and SVHN [LeCun et al., 1998, Netzer et al., 2011]. Where scale falls short, we find tailored learning objectives and training data quality are promising levers for relations and reasoning. Finally, we provide practical recommendations on which models practitioners should select. For example, we find large open models such as Eva ViT-E/14 to be a good choice for a general-purpose VLM while models such as NegCLIP excel at specialized tasks such as visual relations.

To facilitate systematic, comparable, yet easy-to-run evaluations we distill the many benchmarks into a few representative evaluations. We provide the `UniBench` codebase including the 50+ benchmarks with comparisons against all 59 VLMs as well as the distilled set of representative benchmarks that can run in less than 5 minutes on a single A100 GPU. We hope our contribution facilitates thorough and practical evaluation of vision-language model capabilities to faithfully gauge research progress and surface promising strategies to advance VLM research.

## 2 Related Works

### 2.1 Visual Models From Natural Language Supervision

Visual models trained with natural language supervision have revolutionized computer vision by enabling models to learn rich, joint representations of images and text. A seminal work in this area is CLIP (Contrastive Language–Image Pre-training) introduced by Radford et al. [2021b], which demonstrated that pre-training on a large dataset of image-caption pairs using a contrastive objective yields models with remarkable zero-shot transfer capabilities to downstream tasks.

Following CLIP's success, numerous methods have been proposed to enhance visual models through natural language supervision [Bordes et al., 2024, Jia et al., 2021, Yao et al., 2021, Yu et al., 2022b, Li et al., 2022b, Singh et al., 2022b, Gadre et al., 2023a]. These models vary in their approaches, including differences in backbone architectures, training objectives (contrastive learning, image-text matching, masked language modeling), and the scale and quality of the training data. To assess the capabilities of these models, the community has developed a diverse set of benchmarks that test various aspects of visual and multimodal understanding [Yuksekgonul et al., 2023, Thrush et al., 2022b, Hsieh et al., 2024].

However, the proliferation of benchmarks and models has led to a fragmented evaluation landscape, making it challenging to comprehensively assess and compare models. Different benchmarks emphasize different capabilities, and inconsistent evaluation protocols hinder direct comparison. This fragmentation underscores the need for unified evaluation frameworks like `UniBench`, which aim to provide a cohesive and comprehensive suite of benchmarks covering a wide range of vision-language understanding tasks.

## 2.2 CLIP-Style versus LLM-style Evaluation

Evaluation of VLMs has been an active area of research in recent years [Li et al., 2023a, Yue et al., 2024, Liu et al., 2024, Salin et al., 2023, Bitton et al., 2023]. While these benchmarks provide an insightful perspective of VLM capabilities, they primarily focus on LLM-style evaluations, which generate tokens or text as output. These benchmarks are not suitable for evaluating CLIP-like VLMs, which focus on vision-language classification and understanding capabilities. As a result, they do not allow for direct comparisons with CLIP-Style models.

CLIP-Style evaluation is a widely used approach that calculates the similarity between the image representation and text label. This method focuses on vision-language classification and understanding capabilities, making it particularly useful for evaluating models used as backbone/foundation models for image generation and fine-grain visual tasks [Rombach et al., 2021, Ramesh et al., 2021, Saharia et al., 2022]. In contrast, LLM-style evaluation asks the model to demonstrate its knowledge via text generation. While this approach is suitable for evaluating models designed for text-based tasks, it may not be as effective for evaluating models focused on visual tasks. The key difference between CLIP-Style and LLM-style evaluation lies in their respective objectives: CLIP-Style aims to assess a model's ability to align visual and textual representations, whereas LLM-style focuses on assessing a model's ability to generate coherent and accurate text.

`UniBench` focuses on CLIP-Style evaluation, which provides a more comprehensive understanding of a model's visual reasoning capabilities. By concentrating on traditional zero-shot tasks and predefined choices, we enable an apples-to-apples comparison of progress over the past few years, shedding light on promising directions for future research.

## 3 `UniBench`: A comprehensive unified evaluation framework for VLMs

Here we describe the benchmarks, protocols, and axes of progress that comprise `UniBench` as well as the VLMs evaluated.

### 3.1 VLMs Considered in `UniBench`

We evaluate 59 openly available VLMs across a range of model sizes, pre-training dataset sizes, learning objectives, and architectures (full list in Appendix Table 6). For traning dataset size, we include models trained and/or fine-tuned with datasets ranging from 13 million to 12.8 billion samples; including DataComp [Gadre et al., 2023b] (small, medium, large, and extra-large), LIAON [Schuhmann et al., 2022] (400M, 2B, 5B), MetaCLIP [Xu et al., 2023] (400M and 2.5B), Flickr [Young et al., 2014], PMD [Singh et al., 2022a], and COCO [Lin et al., 2015]. For model size and architecture, we categorize models based on the number of parameters and whether these models are convolutional or transformer-based models, ranging from ResNet50 [He et al., 2016] with 38 million parameters to EVA02 ViT E [Fang et al., 2023b] with 4.3 billion parameters.

**Evaluation Procedure** We evaluate performance of zero-shot classification benchmarks similar to [Radford et al., 2021b], by contrasting the representations of class labels (averaged across prompts

as defined by Cherti et al. [2022]) with the image representations and using the class with the highest probability as the predicted class. For relation benchmarks, we follow the standard protocol of contrasting correct and incorrect captions with image representations.

## 3.2 Benchmark Types

To better navigate the overwhelming number of VLM benchmarks, we classify benchmarks into seven distinct types (Figure 1 each covering an key aspect of model performance):

1. **Non-Naural Images:** Consisting of PCam[Veeling et al., 2018], Diabetic Retinopathy[Wang and Yang, 2018], ImageNet Sketch[Wang et al., 2019], imagnetr[Hendrycks et al., 2021a], eurosat[Helber et al., 2019, 2018], and resisc45[Cheng et al., 2017], these benchmarks evaluate the models' ability to handle non-natural images, such as computer-generated graphics, medical images, or satellite imagery.

2. **Object Recognition:** These benchmarks focus on the models' ability to accurately identify and classify objects within images. It includes benchmarks with variety of objects and settings, from everyday items to specific categories like animals or vehicles. Consisting of CUB [Wah et al., 2011], iNaturalist [Van Horn et al., 2018], Pets [Parkhi et al., 2012], MNIST [LeCun et al., 1998], Rendered SST2 [Radford et al., 2021a], SVHN [Netzer et al., 2011], Caltech 101 [Fei-Fei et al., 2004], Stanford Cars [Krause et al., 2013], Cifar 10 [Krizhevsky et al., 2009], Cifar 100 [Krizhevsky et al., 2009], Country211 [Radford et al., 2021a], Dollar Street [Gaviria Rojas et al., 2022], FGVC Aircraft [Maji et al., 2013], Flowers 102 [Nilsback and Zisserman, 2008], Food 101 [Bossard et al., 2014], GTSB [Stallkamp et al., 2012], STL-10 [Coates et al., 2011], VOC 2007 [Everingham et al.], ImageNet [Deng et al., 2009], Places365 [Zhou et al., 2017], sun397 [Xiao et al., 2010], MNIST Fashion [Xiao et al., 2017], and PUG: ImageNet [Bordes et al., 2023].

3. **Reasoning:** These benchmarks test the models' capacity to understand relationships between objects, spatial reasoning, and logical inference based on visual input. The benchmarks consist of CLEVR [Johnson et al., 2017], dmlab [Zhai et al., 2019], DSPR [Matthey et al., 2017], Kitti [Geiger et al., 2012], smallNORB [LeCun et al., 2004], and CountBench [Paiss et al., 2023].

4. **Robustness:** These benchmarks evaluates the models' resilience to adversarial attacks and variations in image data. It includes tests with perturbed images to see how well the models can maintain performance under challenging conditions. For example, the ObjectNet benchmark introduces changes in object position, scale, and background, while the ImageNet-R benchmark focuses on transformations related to many types of image renditions. This collection incdlues ImageNet-E [Li et al., 2023c], ObjectNet [Barbu et al., 2019], ImageNet-A [Hendrycks et al., 2021b], ImageNet-O [Hendrycks et al., 2021b], ImageNet-9 [Xiao et al., 2020], and ImageNet-V2 [Recht et al., 2019].

5. **Relation:** We include relational benchmarks, such as *Visual Genome* [Yuksekgonul et al., 2023], Winoground [Thrush et al., 2022a], and SugarCrepe [Hsieh et al., 2024] designed to evaluate the models' ability to understand and represent relationships between objects within an image, a crucial aspect of visual understanding. For instance, *Visual Genome* benchmark includes a variety of relationships (denoted VG-Relation) and attributions (denoted VG-Attribution) tasks, such as spatial relationships (*e.g*, "above", "next to"), action relationships (*e.g*, "riding", "holding"), and appropriate attribution (*e.g*, "the brown horse and the orange cat" vs. "the orange horse and the orange brown").

6. **Texture:** We rely on DTD [Cimpoi et al., 2014] a benchmark focusing on the models' capability to recognize and differentiate textures within images, which is crucial for tasks such as material recognition and scene understanding.

7. **Corruption:** Consisting of ImageNet-C benchmark [Hendrycks and Dietterich, 2019] introduces various types of image corruptions, such as noise, blur, and digital artifacts. These corruptions simulate the types of degradation that images may undergo in real-world scenarios, such as poor lighting conditions, low-quality cameras, or transmission errors.

## 3.3 Benchmark Capabilities

We further breakdown benchmarks into several capabilities:

1–3. **Depth Estimation, Pose Detection, and Spatial Understanding:** Assessing the models' ability to estimate the depth of objects and scenes from images, and detect object poses which is crucial for understanding spatial relationships.

4–5. **Medical and Satellite:** Testing the models' performance on medical imaging tasks, such as identifying diseases or conditions from medical scans while testing on satellite imagery requires requires recognizing and interpreting land use, terrain, and other geographic features.

6–7. **Counting and Character Recognition:** Assessing the models' ability to identify digits and count objects within images, a fundamental skill for quantitative understanding.

8. **Geographic Diversity:** Evaluating the models' capability to recognize and interpret images from diverse geographic locations and settings.

9. **Scene Recognition:** Measuring how well models can identify and classify different scenes or environments.

10–12. **Standard Object Recognition, ImageNet and Challenging ImageNet:** Evaluating performance on the widely used benchmark for object recognition. We also include the ubiqutous ImageNet and more difficult variants of ImageNet to evaluate model robustness and adaptability.

13. **Specific Classification:** Evaluating models on tasks that require classification of specific categories or fine-grained distinctions between similar objects.

14. **Texture Detection:** Assessing the models' ability to recognize and differentiate various textures within images.

15. **Rendition:** Assessing models' performance on tasks involving rendered or synthetic images, which differ from natural photographs.

16–17. **Corruptions and Natural Transformations:** Evaluating robustness to image corruptions, such as noise, blur, and other artifacts that degrade image quality whereas natural transformations includes common changes in lighting, rotation, or perspective.

### 3.4 `UniBench`: a systematic, practical VLM evaluation

UniBench is framework for comprehensive, fast, and easy-to-use evaluation of VLMs. UniBench also has the ability to expand the existing set of benchmarks and VLMs, as shown in (Code Snippet 1).

```python
import unibench
from unibench.models_zoo.wrappers.clip import ClipModel
from torchvision.datasets import FashionMNIST

evaluator = unibench.Evaluator()
model = partial(
            ClipModel,
            model=model,
            model_name="vitamin_l_comp1b",
            tokenizer=tokenizer,
            input_resolution=model.visual.image_size[0],
            logit_scale=model.logit_scale,
        )
evaluator.add_model(model=model)
class_names = ["T-shirt/top",...]
templates = ["an image of {}", ...]
benchmark = partial(FashionMNIST, root="./", train=False, download=
    True)
handler = partial(ZeroShotBenchmarkHandler, benchmark_name="
    fashion_mnist_new", classes=class_names,templates=templates)
evaluator.add_benchmark(benchmark, handler, meta_data={"
    benchmark_type": "object recognition"})
evaluator.evaluate()
```

Code Snippet 1: Running UniBench with a custom model and a new benchmark. UniBench accepts any torchvision dataset type.

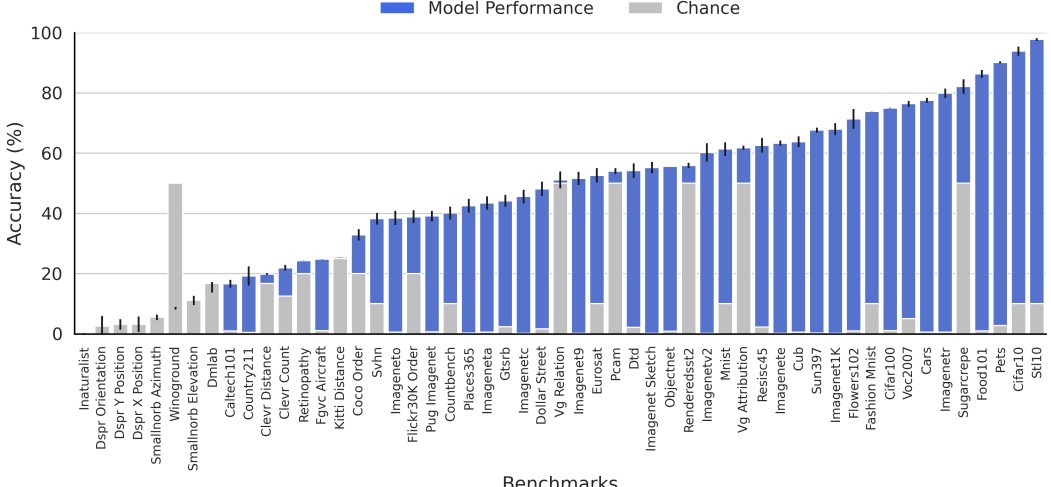

Figure 2: **Median performance of all 59 VLMs on 53 benchmarks, illustrating that despite advancements, VLMs still struggle on several benchmarks.** Benchmarks that barely exceed chance-level performance include Winoground, iNaturalist, DSPR, Small Norb, dmlab, Clevr, PCam, Renderedssst2, and Kitti. Blue bars represent the median zero-shot performance of the models, while grey bars indicate the chance-level for each benchmark.

## 4   Gauging progress in Vision Language Modeling with `UniBench`

We show the overall median performance of the nearly 60 VLMs we examined on 53 benchmarks in Figure 2 ranked by their zero-shot classification performance. The results suggest that, while VLMs perform remarkably well on many tasks, for others, VLM performance is near or below random chance level. These results highlight the need for a unifying pipeline to systematically surface model limitations.

### 4.1   Scaling improves many benchmarks, but offers little benefit for reasoning and relations

**Scaling training dataset size hardly helps for reasoning and relations.**   While scaling training dataset size improves performance across many tasks, this trend does not hold for benchmarks assessing relation understanding and reasoning capabilities. To control for other confounding factors, we fix the architecture, learning paradigm, model size (for right panel), and training dataset size (for left) by using the same CLIP ViT-B/32 model and LAION 400M dataset, respectively Figure 3. The results suggest despite increasing the training dataset size by a factor of $1000\times$, relational and reasoning benchmarks performance is fairly flat compared to the significant boost in performance on other tasks. We observe a similar trend overall when we include all 59 models in Appendix Figure 7. We specifically pinpoint capabilities such as Depth Estimation, Spatial Understanding, Counting, Scene and Text Recognition, as the underlying capabilities where scale does not lead to improvements as shown in Figure 4.

**Scaling model size also offers little to no benefit for reasoning or relations.**   When we scale models' size from 86 million parameters to 1 billion parameters, we also find that models struggle to scale on similar proportions on relation and reasoning tasks as shown in Figure 3. While for other benchmark types including object recognition, robustness, etc. performance improves by 17.4% as model size scales by $11\times$, relations and reasoning improve by a modest 3.41% with a fairly flat scaling curve. Similar to scaling training dataset size, scaling model size also offers little benefit for capabilities such as Depth Estimation, Spatial Understanding, Counting, Scene and Text Recognition as shown in Figure 4.

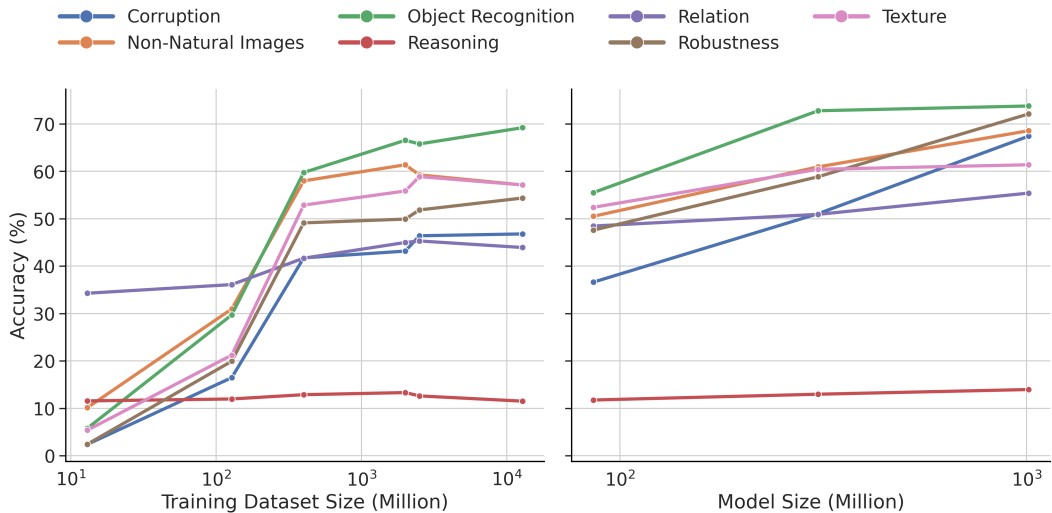

Figure 3: **The effect of scaling model and training dataset size using a fixed architecture and learning paradigm.** Zero-shot performance of models on various benchmark types. We investigate the impact of training dataset size (left), and model size on various benchmark types (right). To isolate the effect of scale, we fix the architecture, learning paradigm, model size (for right panel), and training dataset size (for left) by using the same CLIP ViT-B/32 model and LAION 400M dataset, respectively. We observe a similar trend when measured across all 59 models as shown in Appendix Figure 7

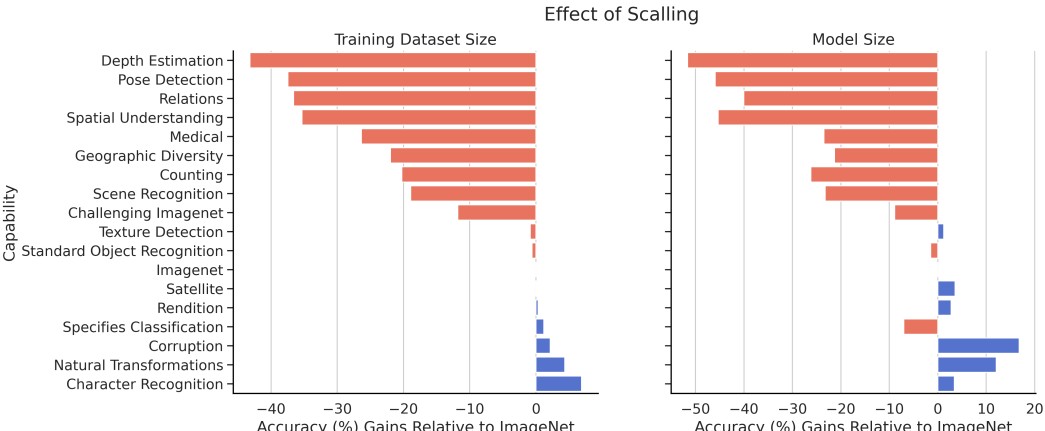

Figure 4: **The effect scaling of training dataset (left) and model size (right) across capabilities for all models.** Accuracy is the difference in performance between the most scaled and the least scaled model across capabilities relative to ImageNet performance.

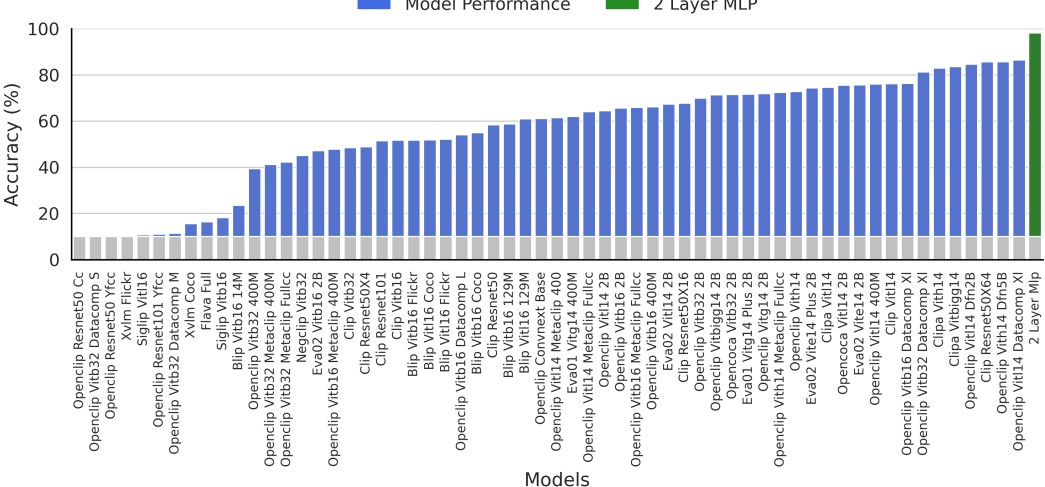

Figure 5: **Performance of 59 VLMs on MNIST, showing despite progress, VLMs still struggle on MNIST**. Blue bars represent zero-shot performance of models, grey bars represent the chance-level for MNIST, and green bar shows performance for a 2-Layer MLP.

## 4.2 A Case Study: Digit Recognition and Counting are notable limitations for VLMs even with the right training data

A surprising aspect of VLMs is their poor performance on benchmarks that are traditionally considered straightforward, such as MNIST, CIFAR-10, and CIFAR-100, as shown in Figure 2. For example, a simple 2-layer MLP achieves 99% accuracy on MNIST [Wan et al., 2013] significantly outperforming all 59 VLMs we studied. To delve deeper into this unexpected result, we controlled for several variables:

1. **VLM confusions go beyond top-1:** To further understand the performance results on MNIST, we compute more generous top-2,-3,-4, and -5 accuracy measures to understand whether models confuse similar digits. We show in Appendix Figure 10 that even when we compute top-5 accuracy (with 50% being chance), VLMs barely reach 90% accuracy suggesting poor performance is not due to minor confusions among digits.

2. **Prompt engineering isn't enough for good performance:** To ensure that the poor performance was not an artifact of the prompts used, we tested multiple hand-crafted prompts that included detailed descriptions of the image characteristics Appendix Figure 9. Despite these tailored prompts, which explicitly described the black-and-white nature and content of the images, the performance still lagged significantly behind simpler models.

3. **Training data contains ample samples with digit concept:** We investigated whether the subpar performance could be attributed to a lack of training images containing digit concepts by analyzing the popular LAION 400M dataset. Our findings reveal a substantial number of captions with both word digits (100k-2M) and integer digits (15M-48M) in the training captions, suggesting that the poor performance is not merely due to insufficient training data (see Figure 11 for exact counts by digit).

4. **VLMs struggle on other digit benchmarks:** To further explore whether the poor performance on MNIST is indicative of broader issues in number comprehension, we extend our investigation to other benchmarks such as SVHN, CountBench, and ClevrCount (Appendix Figure 6). We find across all benchmarks VLMs struggled with number recognition and counting tasks.

**Takeaway** Despite training on vast datasets, even leading VLMs can struggle with simple tasks solved trivially by much smaller models, including tasks involving basic number comprehension. These findings highlight the need for a comprehensive evaluation pipeline that includes so called, simpler benchmarks, to uncover VLM limitations.

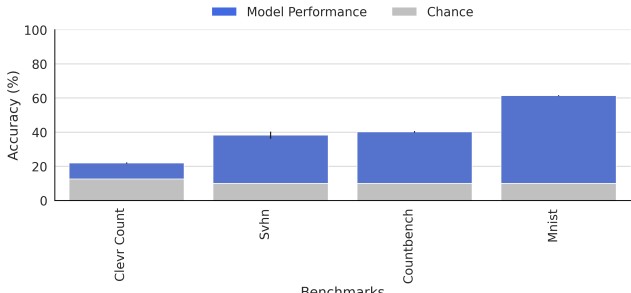

Figure 6: **Median performance of 59 VLMs on counting and character recognition benchmarks, showing MNIST performance is not an isolated instance and VLMs generally sruggle with these tasks**. Blue bars represent the median zero-shot performance of models and gray bars represent random chance-level.

## 4.3    What contributes to better model performance?

We have shown both the promise and limitations of scale for VLM performance. We now examine what other levers can overcome the limitations of scale. In particular, we examine other promising factors, such as data quality and learning objectives for improving relational understanding and reasoning.

**Data quality matters more than data quantity.**    Among the 59 VLMs we evaluated, there are models trained from 12.8 million samples to 12.8 billion samples. While the quantity of data is often highlighted as a key driver for improving model performance, the quality of the data can be even more critical. For instance, among all models in Appendix Tables 1 and 5, the top performing models are generally the ones trained on 2 billion samples, which use more strict CLIP score filtering as described in Gadre et al. [2024]. This observation suggests that beyond a certain threshold, simply adding more data does not necessarily translate to better performance. Instead, the composition and quality of the data set become paramount. Models trained on such data are better equipped to generalize from their training environments to real-world applications, demonstrating that strategic curation of data can be more valuable than the sheer volume of data collected.

**Tailored learning objectives can help where scale does not.**    The learning objectives defined during model training phase are pivotal in steering the model's learning process and ultimately its performance on various tasks.  A notable example is NegCLIP [Yuksekgonul et al., 2022], with a tailored learning objective for capturing relations via hard-negatives seems to substantially aid NegCLIP's performance on relational understanding (Appendix Tables 1 and 5). As shown in the original paper NegCLIP's performance isn't simply the result of finetuning with additional data (see Table 6 of Yuksekgonul et al. [2022]), but is thanks to a tailored learning objective involving hard negatives. NegCLIP, with only 86M parameters, significantly outperforms models up to $50\times$ larger with an overall performance of 70.4%, compared to only 50.5% for the largest EVA ViT-E/14 model with 4.3B parameters. Similarly, Paiss et al. [2023] tailored learning objective for VLMs can have significantly improve performance on counting tasks.

## 4.4    Which model should I use?

Finally, we provide recommendations for practitioners to select the most suitable openly available VLM. For an overall high-performing model across the axes we measured, models with large ViT encoders trained on large datasets exhibit the highest overall performance, with Eva-2 ViT-E/14 leading the way. For relations, counting, or related capabilities, we rank the top and worst performing models in Appendix Table 5.

| Benchmark Type | Mean Performance | Top | | Top vs Worst Scale | | Worst | |
|---|---|---|---|---|---|---|---|
| | | Model | Performance | Training Dataset Size | Model Size | Performance | Model |
| Corruption | 46.2 | EVA02 ViT E 14 | 74.3 | 153× | 50× | 2.4 | DataComp ViT B 32 |
| Non-Natural Images | 54.1 | EVA02 ViT E 14 | 74.6 | 153× | 50× | 16.1 | DataComp ViT B 32 |
| Object Recognition | 55.0 | CLIPA ViT G 14 | 71.1 | 98× | 21× | 12.1 | DataComp ViT B 32 |
| Reasoning | 14.9 | OpenCLIP ViT g 14 | 19.0 | 133× | 18× | 10.6 | OpenCLIP ResNet101 |
| Relation | 46.7 | NegCLIP ViT B 32 | 66.8 | 30× | 1× | 33.2 | DataComp ViT B 32 |
| Robustness | 52.1 | EVA02 ViT E 14 | 72.8 | 153× | 50× | 3.8 | DataComp ViT B 32 |
| Texture | 53.5 | MetaCLIP ViT H 14 | 72.5 | 192× | 7× | 5.4 | DataComp ViT B 32 |
| Overall | 46.1 | EVA02 ViT E 14 | 61.2 | 153× | 50× | 12.1 | DataComp ViT B 32 |

Table 1: List of all evaluated benchmark types with their corresponding mean performance across models, the best and worst performing models. The Top vs. Worst Scale shows the proportion difference between the worst and best model on the training dataset size and the model size.

# 5   `UniBench`: A Practical Way Forward for Faster Comprehensive VLM Evaluations

While ideally, evaluating VLMs across all 53 benchmarks would provide the most comprehensive insights, the computational demands and complexity of parsing such extensive data can be overwhelming (6 million images to evaluate; 2+ hours for one model on an A100 GPU). While ImageNet maybe a tempting candidate as it correlates with many benchmarks, for many others, specifically 18 of the 53 benchmarks, ImageNet performance is poorly or negatively correlated Appendix Figure 12. This suggests that success on ImageNet does not universally translate to proficiency in all tasks.

**Comprehensive VLM evaluation with `UniBench` in 5 minutes.**   To streamline evaluation, we distill the full set of benchmarks in `UniBench` into seven benchmarks most representative of each axis of progress (via correlations in Appendix A.6). Fortunately, in `UniBench` this comprehensive set of benchmarks runs in 5 minutes on a single A100 GPU (for ViT-B/32), offering a fast, yet comprehensive evaluation pipeline.

# 6   Discussion

**Limitations**   While we invested a considerable effort to include as comprehensive set of models and benchmarks as possible, there of course will always be new ones we do not cover. We focus especially on vision-centric benchmarks to track progress since the early contrastive vision-language models. To mitigate that, we provide a flexible interfaces to extend `UniBench` with additional models or benchmarks (see code 1). Our analysis is also limited to the standard zero-shot evaluation protocol.

**Impact**   To guide the research community in navigating the overwhelming and fragmented landscape of VLM benchmarks, we introduced `UniBench`. `UniBench` provides a unified implementation of 50+ benchmarks, out-of-the-box comparisons across nearly 60 open VLMs, and a distilled fast-to-run set of representative benchmarks that can run on in 5 minutes a single GPU. In doing so, we uncover the limits of scale for reasoning and relations, the promise of data quality and tailored learning objectives, as well as offer recommendations for which VLMs practitioners should use. We hope `UniBench` is a step towards avoiding the blindspots in VLM evaluations, enabling researchers to comprehensively, yet efficiently evaluate progress.

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

# A  Appendix

## A.1  `UniBench` **Implementation Details**

We have developed `UniBench` to be easy-to-run library to allow researchers to systematically compare and contrast exsisting (n=59 ) and new VLMs on 53 benchmarks. To evaluate new VLMs that expand beyond the already implemented 59 VLMs, users need to follow Code Snippet 2. Users would need to create a class that inherent from `ClipModel` from `uni_bench.models_zoo` with `get_image_embeddings` and `get_text_embeddings` methods implemented. `get_image_embeddings` and `get_text_embeddings` methods takes images and captions as input, respectively, and returns a tensor of encoded representations.

```python
from unibench.models_zoo import ClipModel
import unibench

class CustomModel(ClipModel):

    @torch.no_grad()
    # Output tensor of final layer of image encoder
    def get_image_embeddings(self, images):
        ...

    @torch.no_grad()
    # Output tensor of final layer of text encoder given captions
    def get_text_embeddings(self, captions):
        ...

evaluator = unibench.Evaluator() # Initialize Evaluator class
new_model = CustomModel() # Initialize new model
evaluator.add_model(new_model) # add new model to the evaluation
    pipeline
evaluator.evaluate() # run the evaluation
```

Code Snippet 2: Custom Model Example

## A.2  Natural Language Output Models on UniBench

As described in Section 2.2, LLM-style models defined as models that generate tokens/text as output. Thereby, making them hard to compare with CLIP-style VLMs. In UniBench, we also incorporated LLM-style models in a control experiments. While, LLM-style benchmarks are not suitable for evaluating CLIP-like VLMs, benchmarks in UniBench are capable of testing both LLM and CLIP style models. Following Matsuura et al. [2023] methodology, we evaluated Llava 1.5 [Liu et al., 2023] - a LLM-style VLM - on various benchmark types in UniBench (Table 2). In Table 2, we evaluated 7 and 13 billion scales of Llava.

| Model Name | Corruption | Non-natural Images | Object Recognition | Reasoning | Relation | Robustness | Texture |
|---|---|---|---|---|---|---|---|
| Llava 1.5 13B [Liu et al., 2023] | 31 | 50 | 36 | 11 | 41 | 24 | 34 |
| Llava 1.5 7B [Liu et al., 2023] | 29 | 51 | 32 | 12 | 42 | 23 | 28 |

Table 2: Performance (%) of Llava 1.5 on different Benchmark types.

## A.3  Gauging progress in Vision Language Models

**Scaling improves many benchmarks, but offers little benefit for reasoning and relation.**  Appendix Figure 7 shows that despite increasing the training dataset size by a factor of $1000\times$ and model size by a factor of $11\times$, relational and reasoning benchmarks performance is fairly flat compared to the significant boost in performance on other tasks. We further pinpoint capabilities such as Depth Estimation, Spatial Understanding, Counting, Scene and Text Recognition, as the underlying capabilities where scale does not lead to improvements as shown in Figure 8.

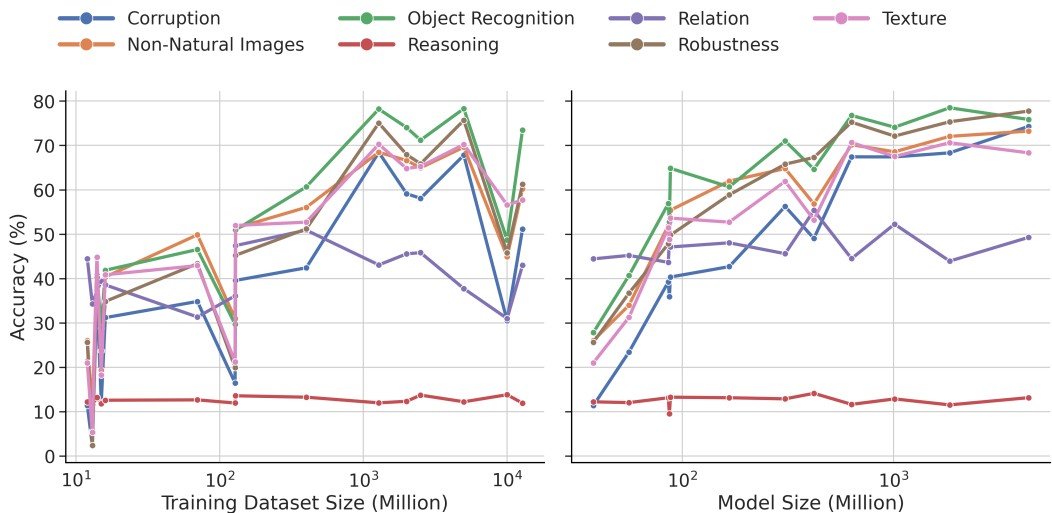

Figure 7: **The effect of scaling model and training dataset size on all models.** Median zero-shot performance of models on various benchmark types. We investigate the impact of training dataset size (left), and model size on various benchmark types (right).

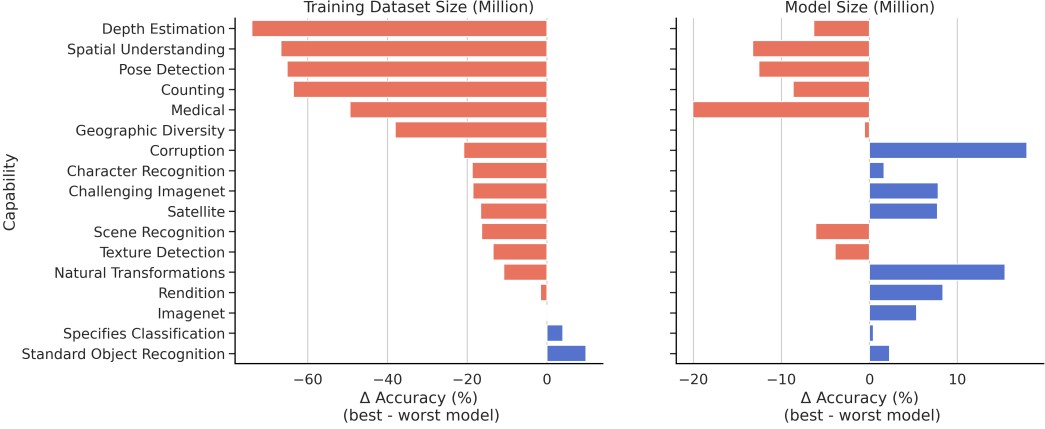

Figure 8: **Benchmark capabilities performance does not scale with dataset and model size** Median zero-shot performance of models on various benchmark capabilities. We investigate the impact of dataset size (left), and model size on various benchmark capabilities (right). We isolate the effect of training data size keeping other factors such as architecture, learning objective, and model size fixed only using ViT B32 (left). For right panel subfigure, we isolate the effect of model size keeping other factors such as architecture, learning objective, and training data size fixed only using LIAON 400M (right).

## A.4 Impact of Prompts on MNIST Performance

The MNIST benchmark, featuring handwritten digits, was subjected to various prompting strategies to evaluate their impact on model performance. Our findings reveal a distinct hierarchy in performance based on the type of prompts used. The benchmark was tested with both numeral formats ("zero-nine" and "0-9") and different prompt styles (specialized word prompts, specialized digit prompts, and a basic prompt) (Figure 9).

### A.4.1 Hierarchy of Prompt Performance

The performance of the MNIST model varied significantly across different prompt types and formats, arranged here from best to worst performing setups: 1. Word digits ("zero-nine") with specialized word prompts 2. Word digits ("zero-nine") with basic prompt 3. Word digits ("zero-nine") with specialized digit prompts 4. Digits ("0-9") with specialized digit prompts 5. Digits ("0-9") with basic prompt 6. Digits ("0-9") with specialized word prompts

### A.4.2 Specialized Word Prompts

These prompts provided detailed descriptions and contexts, significantly enhancing the model's ability to recognize and interpret the digits accurately. Examples include:

- "showcasing the digit {}, is this image."
- "this number {} is represented in a handwritten form."
- "the numeral {} is captured in this snapshot."
- "the digit {} is depicted visually in this image."
- "this image is a graphical representation of the number {}."
- "this is an illustration of the digit {}."
- "this image represents the digit {} in a handwritten form."
- "the number {} is sketched as a digit in this image."
- "this is a photograph of the digit {}."
- "the number {} is drawn as a digit in this image."

### A.4.3 Specialized Digit Prompts

These prompts explicitly mention the format or style of the digit, aiding in recognition but to a lesser extent compared to specialized word prompts. Examples include:

- "A photo of the number: '{}'."
- "A digit drawing of the number: '{}'."
- "A digit sketch of the number: '{}'."
- "A handwritten digit image of: '{}'."
- "A digit illustration of: '{}'."
- "A graphical representation of the number: '{}'."
- "A visual depiction of the digit: '{}'."
- "A snapshot of the numeral: '{}'."
- "A handwritten representation of the number: '{}'."
- "An image showcasing the digit: '{}'."

### A.4.4 Basic Prompt

The basic prompt used:

- "a photo of the number: '{}'."

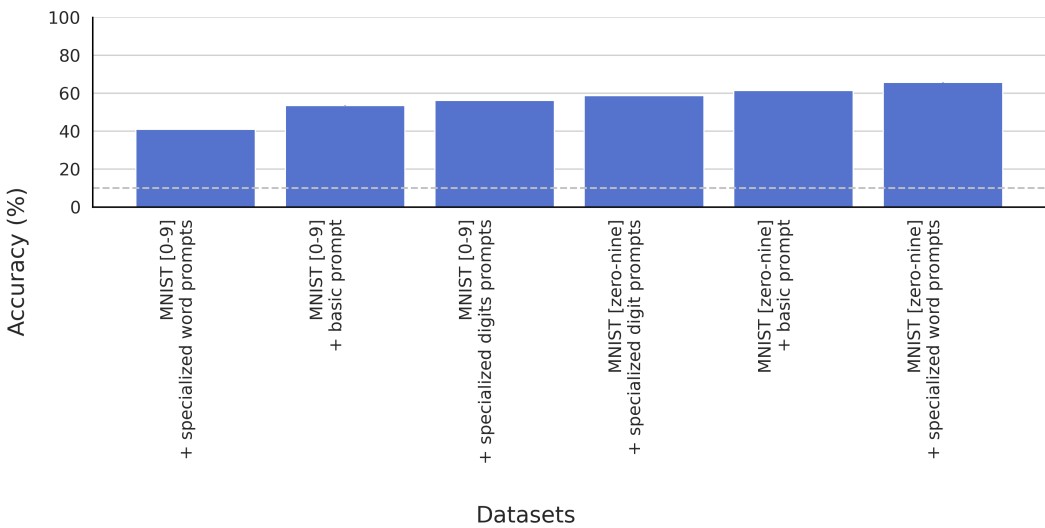

Figure 9: **Median performance of 59 VLMs on MNIST while varying prompts and labels**. Blue bars represent the median zero-shot performance of models and dashed-grey line represents the chance-level for MNIST.

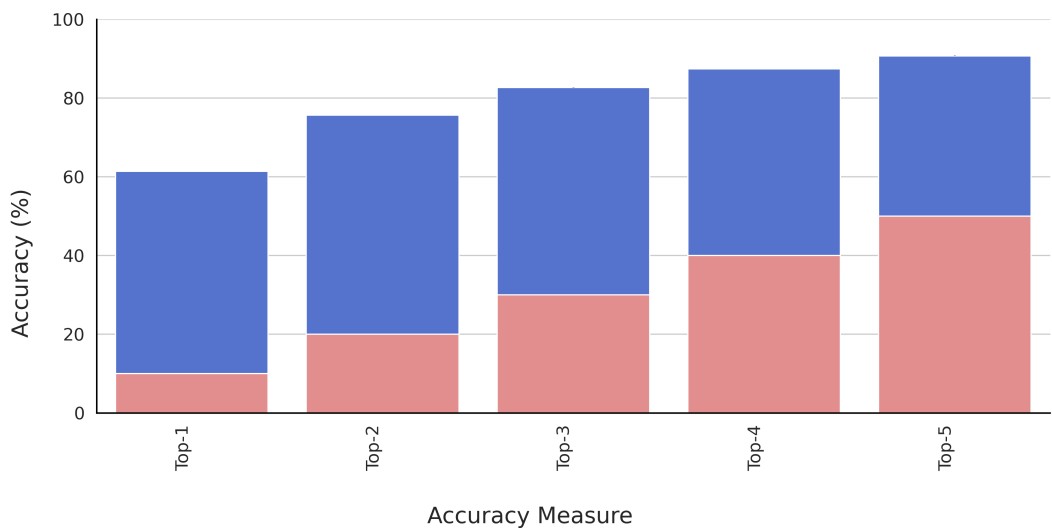

Figure 10: **Median performance of 59 VLMs on MNIST while varying accuracy measure from top-1 to top-5**. The following further shows that VLMs' performance on MNIST is not due mismatch between top-1 and top-5 guesses. Blue bars represent the median zero-shot performance of models and red bars represents the chance-level for benchmarks.

This structured analysis clearly demonstrates how the specificity and relevance of the prompt significantly influence the performance of VLMs. We investigated whether the subpar performance could be attributed to a lack of training images containing digit concepts by analyzing the popular LAION 400M dataset. Our findings reveal a substantial number of captions with both word digits (100k-2M) and integer digits (15M-48M) in the training captions, suggesting that the poor performance is not merely due to insufficient training data (see Figure 11 for exact counts by digit). To further understand the performance results on MNIST, we compute more generous top-2,-3,-4, and -5 accuracy measures to understand whether models confuse similar digits. We show in Appendix Figure 10 that even when we compute top-5 accuracy (with 50% being chance), VLMs barely reach 90% accuracy suggesting poor performance is not due to minor confusions among digits.

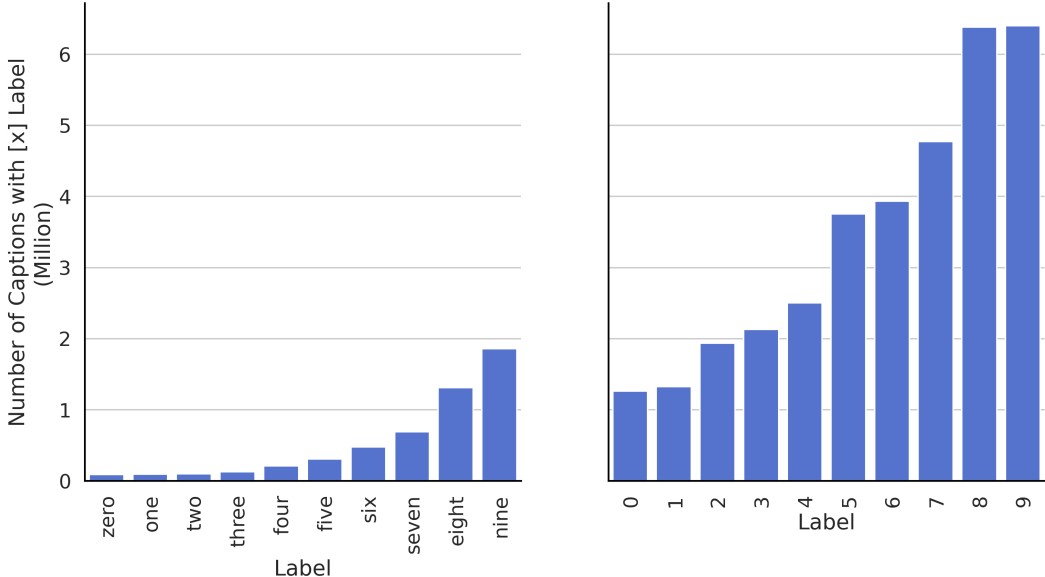

Figure 11: **Frequency of different digits in LAION-400M, showing substantial frequency of digits in visual diet of VLMs.** Left panel counts the number of words of the digits i.e. [zero-nine] and right panel counts the number of digits in LAION-400M.

### A.5 Correlation of ImageNet with Other Benchmarks

ImageNet, often considered a cornerstone in the field of computer vision, has been widely used as a benchmark to evaluate the performance of image recognition models. Its extensive dataset and challenging classification tasks have set a standard for algorithm development and comparison. However, while ImageNet correlates well with many benchmarks, it does not exhibit a universal correlation across all tasks. Our analysis reveals that for a significant number of benchmarks, specifically 18 out of the 53 benchmarks analyzed, the performance on ImageNet is poorly or negatively correlated. This is illustrated in Appendix Figure 12, which provides a detailed comparison of benchmark performances. This finding suggests that success on ImageNet does not necessarily translate to proficiency in all visual tasks.

### A.6 A Practical Subset of Benchmarks

While ideally, evaluating VLMs across all 53 benchmarks would provide the most comprehensive insights, the computational demands and complexity of parsing such extensive data can be overwhelming (6 million images to evaluate; 2+ hours for one model on an A100 GPU). To streamline evaluation, we distill the full set of benchmarks in `UniBench` into seven benchmark types and 17 capabilities. These categorizations are based on benchmarks that correlate strongly with other benchmarks within each benchmark type and capability (Tables 3 and 4).

### A.7 Weighted Average Performance

To account for the varying difficulties across tasks, we compute the weighted average performance of each model by normalizing their scores relative to the performance of CLIP B/32. We use CLIP B/32 as a baseline because its performance effectively captures the inherent complexity of each task, serving as a proxy for task difficulty.

Figure 13 illustrates the normalization results in lower overall performance scores for all models. However, it does not affect the relative rankings among them. This consistency suggests that while task difficulty impacts absolute performance metrics, the comparative effectiveness of the models remains stable across different levels of task complexity.

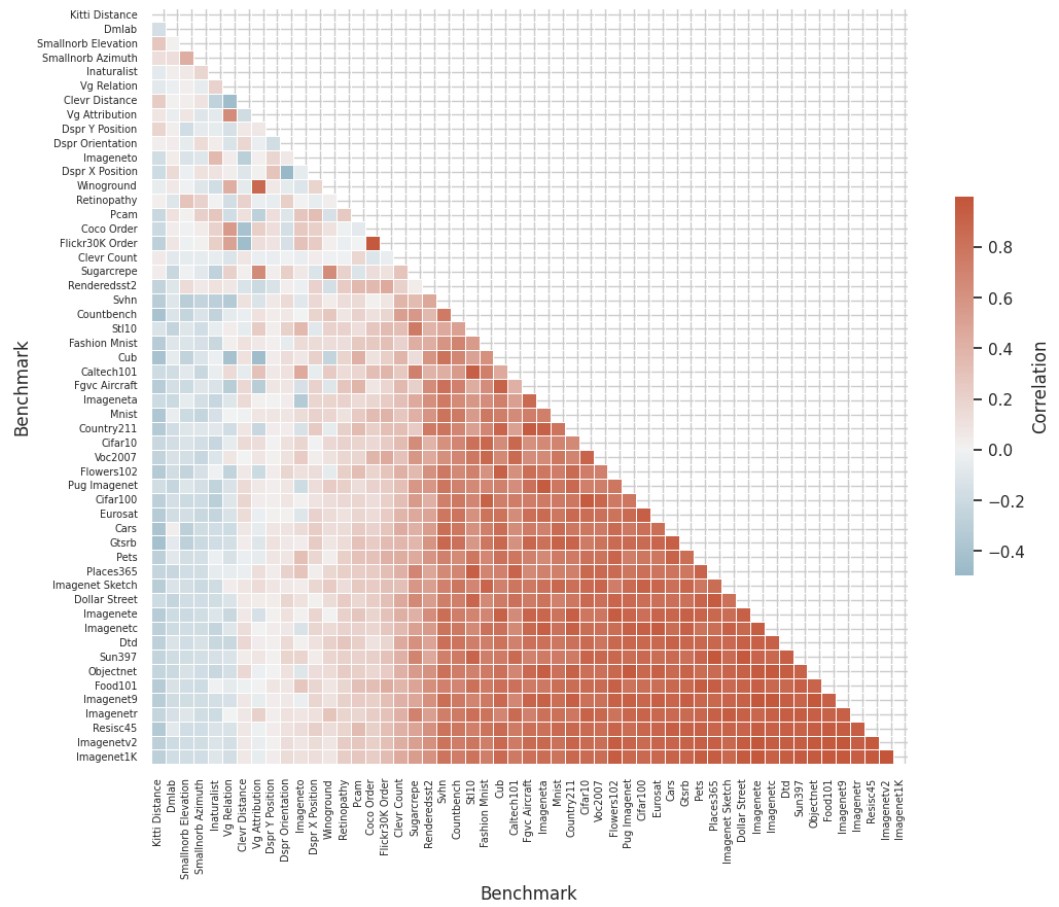

Figure 12: **Correlation matrix of models' performance across all benchmarks.**

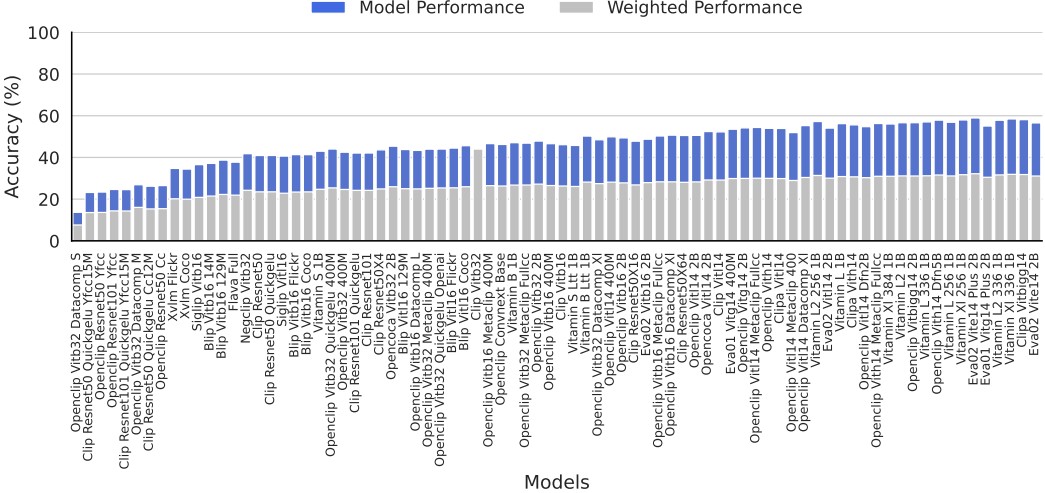

Figure 13: **Weighted Average Performance** for each model using CLIP B/32 as the baseline model performance (as a proxy for task difficulty

| Benchmark Type | Most Correlated Benchmark | Correlation Value |
|---|---|---|
| Object recognition | ImageNet-1k | 0.82 |
| Reasoning (Counting) | CountBench | 0.76 |
| Reasoning (Spatial) | DSPR Position | 0.29 |
| Relation | VG Attribution | 0.57 |
| Texture | DTD | 1 |
| Non-Natural Images | Resisc45 | 0.72 |
| Robustness | ImageNet-v2 | 0.81 |
| Corruption | ImageNet-c | 1 |

Table 3: **Evaluate on a curated list of benchmark types, rather than the full set, to save time.** The list includes benchmarks that correlate strongly with other benchmarks for each benchmark type.

| Capabilities | Most Correlated Benchmark | Correlation Value |
|---|---|---|
| standard object recognition | food101 | 0.85 |
| counting | countbench | 0.76 |
| spatial understanding | dspr y position | 0.29 |
| relations | vg attribution | 0.57 |
| geographic diversity | dollar street | 0.89 |
| specifies classification | flowers102 | 0.7 |
| depth estimation | dmlab | 0.42 |
| pose detection | smallnorb azimuth | 0.57 |
| texture detection | dtd | 1 |
| satellite | eurosat | 0.95 |
| character recognition | mnist | 0.88 |
| imagenet | imagenet1k | 1 |
| natural transformations | imagenet9 | 0.99 |
| rendition | imagenetr | 0.97 |
| challenging imagenet | imagenetv2 | 0.65 |
| corruption | imagenetc | 1 |
| medical | retinopathy | 0.64 |
| scene recognition | sun397 | 0.99 |

Table 4: **Evaluate on a curated list of capabilities, rather than the full set, to save time.** The list includes benchmarks that correlate strongly with other benchmarks for each capability.

| Benchmark Type | Mean Performance | Top | | Top vs Worst Scale | | Worst | |
|---|---|---|---|---|---|---|---|
| | | Model | Performance | Training Dataset Size | Model Size | Performance | Model |
| Challenging Imagenet | 47.8 | EVA02 ViT E 14 | 64.4 | 153 | 50 | 5.0 | DataComp ViT B 32 |
| Character Recognition | 54.8 | CLIPA ViT G 14 | 74.3 | 85 | 48 | 20.5 | OpenCLIP ResNet50 |
| Corruption | 46.1 | EVA02 ViT E 14 | 74.3 | 153 | 50 | 2.3 | DataComp ViT B 32 |
| Counting | 31.4 | OpenCOCA ViT L 14 | 53.1 | 153 | 3 | 11.5 | DataComp ViT B 32 |
| Depth Estimation | 20.4 | DataComp ViT B 16 | 27.6 | 0.6 | 0.1 | 12.4 | OpenCLIP ViT H 14 |
| Geographic Diversity | 33.8 | CLIPA ViT G 14 | 46.8 | 98 | 21 | 5.3 | DataComp ViT B 32 |
| Imagenet | 65.7 | OpenCLIP ViT H 14 | 83.1 | 384 | 7 | 3.9 | DataComp ViT B 32 |
| Medical | 43.3 | MetaCLIP ViT L 14 | 68.6 | 0.3 | 3 | 26.8 | DataComp ViT B 16 |
| Natural Transformations | 56.2 | CLIPA ViT G 14 | 81.7 | 98 | 21 | 2.5 | DataComp ViT B 32 |
| Pose Detection | 3.9 | OpenCLIP ViT B 32 | 4.7 | 5 | 0.9 | 3.3 | OpenCLIP ConvNext |
| Relations | 46.7 | NegCLIP ViT B 32 | 66.7 | 30 | 1 | 33.2 | DataComp ViT B 32 |
| Rendition | 63.7 | CLIPA ViT G 14 | 84.2 | 98 | 21 | 3.8 | DataComp ViT B 32 |
| Satellite | 55.2 | EVA02 ViT E 14 | 75.7 | 153 | 50 | 12.3 | DataComp ViT B 32 |
| Scene Recognition | 53.0 | OpenCLIP ViT H 14 | 61.7 | 384 | 7 | 6.3 | DataComp ViT B 32 |
| Spatial Understanding | 9.1 | MetaCLIP ViT L 14 | 11.3 | 1 | 3 | 6.3 | CLIP ResNet50x4 |
| Specifies Classification | 51.7 | OpenCLIP ViT H 14 | 68.9 | 384 | 7 | 2.8 | DataComp ViT B 32 |
| Standard Object Recognition | 60.0 | CLIPA ViT G 14 | 77.1 | 98 | 21 | 13.8 | DataComp ViT B 32 |
| Texture Detection | 53.4 | MetaCLIP ViT H 14 | 72.4 | 192 | 7 | 5.3 | DataComp ViT B 32 |
| Overall | 44.2 | EVA02 ViT E 14 | 58.0 | 153 | 50 | 11.3 | DataComp ViT B 32 |

Table 5: List of all evaluated capabilities with their corresponding mean performance across models, the best and the worst performing models. The Top vs. Worst Scale shows the proportion difference between the worst and best model on the training dataset size and the model size.

| Model | Dataset size | Model size | Learning objective | Architecture | Model name |
|---|---|---|---|---|---|
| blip_vitB16_14m Li et al. [2022a] | 14 | 86 | BLIP | vit | BLIP ViT B 16 |
| blip_vitL16_129m Li et al. [2022a] | 129 | 307 | BLIP | vit | BLIP ViT L 16 |
| blip_vitB16_129m Li et al. [2022a] | 129 | 86 | BLIP | vit | BLIP ViT B 16 |
| blip_vitB16_coco Li et al. [2022a] | 129 | 86 | BLIP | vit | BLIP ViT B 16 |
| blip_vitB16_flickr Li et al. [2022a] | 129 | 86 | BLIP | vit | BLIP ViT B 16 |
| blip_vitL16_coco Li et al. [2022a] | 129 | 307 | BLIP | vit | BLIP ViT L 16 |
| blip_vitL16_flickr Li et al. [2022a] | 129 | 307 | BLIP | vit | BLIP ViT L 16 |
| eva02_vitE14_plus_2b Fang et al. [2023b] | 2000 | 4350 | Pure Contrastive | vit | EVA02 ViT E 14 |
| eva02_vitE14_2b Fang et al. [2023b] | 2000 | 4350 | Pure Contrastive | vit | EVA02 ViT E 14 |
| eva02_vitL14_2b Fang et al. [2023b] | 2000 | 307 | Pure Contrastive | vit | EVA02 ViT L 14 |
| eva02_vitB16_2b Fang et al. [2023b] | 2000 | 86 | Pure Contrastive | vit | EVA02 ViT B 16 |
| eva01_vitG14_plus_2b Fang et al. [2022] | 2000 | 1011 | Pure Contrastive | vit | EVA01 ViT g 14 |
| eva01_vitG14_400m Fang et al. [2022] | 400 | 1011 | Pure Contrastive | vit | EVA01 ViT g 14 |
| clipa_vitbigG14 Li et al. [2023b] | 1280 | 1843 | Pure Contrastive | vit | CLIPA ViT G 14 |
| clipa_vitH14 Li et al. [2023b] | 1280 | 633 | Pure Contrastive | vit | CLIPA ViT H 14 |
| clipa_vitL14 Li et al. [2023b] | 1280 | 307 | Pure Contrastive | vit | CLIPA ViT L 14 |
| siglip_vitL16 Zhai et al. [2023] | 10000 | 307 | Contrastive (sigmoid) | vit | SigLIP ViT L 16 |
| siglip_vitB16 Zhai et al. [2023] | 10000 | 86 | Contrastive (sigmoid) | vit | SigLIP ViT B 16 |
| openclip_vitB32_metaclip_fullcc Xu et al. [2023] | 2500 | 86 | Pure Contrastive | vit | MetaCLIP ViT B 32 |
| openclip_vitB16_metaclip_400m Xu et al. [2023] | 400 | 86 | Pure Contrastive | vit | MetaCLIP ViT B 16 |
| openclip_vitB32_metaclip_400m Xu et al. [2023] | 400 | 86 | Pure Contrastive | vit | MetaCLIP ViT B 32 |
| openclip_vitB16_metaclip_fullcc Xu et al. [2023] | 2500 | 86 | Pure Contrastive | vit | MetaCLIP ViT B 16 |
| openclip_vitL14_dfn2b Fang et al. [2023a] | 2000 | 307 | Pure Contrastive | vit | OpenCLIP ViT L 14 |
| openclip_vitL14_metaclip_400 Xu et al. [2023] | 400 | 307 | Pure Contrastive | vit | MetaCLIP ViT L 14 |
| openclip_vitL14_metaclip_fullcc Xu et al. [2023] | 2500 | 307 | Pure Contrastive | vit | MetaCLIP ViT L 14 |
| openclip_vitH14_metaclip_fullcc Xu et al. [2023] | 2500 | 633 | Pure Contrastive | vit | MetaCLIP ViT H 14 |
| openclip_vitH14_dfn5b Fang et al. [2023a] | 5000 | 633 | Pure Contrastive | vit | OpenCLIP ViT H 14 |
| openclip_convnext_base Ilharco et al. [2021] | 400 | 88 | Pure Contrastive | conv | OpenCLIP ConvNext |
| openclip_vitB32_datacomp_s Gadre et al. [2023b] | 13 | 86 | Pure Contrastive | vit | DataComp ViT B 32 |
| openclip_vitB32_datacomp_m Gadre et al. [2023b] | 128 | 86 | Pure Contrastive | vit | DataComp ViT B 32 |
| openclip_vitB32_datacomp_xl Gadre et al. [2023b] | 12800 | 86 | Pure Contrastive | vit | DataComp ViT B 32 |
| openclip_vitB16_datacomp_xl Gadre et al. [2023b] | 12800 | 86 | Pure Contrastive | vit | DataComp ViT B 16 |
| openclip_vitB16_datacomp_l Gadre et al. [2023b] | 1280 | 86 | Pure Contrastive | vit | DataComp ViT B 16 |
| openclip_vitH14 Ilharco et al. [2021] | 2000 | 633 | Pure Contrastive | vit | OpenCLIP ViT H 14 |
| xvlm_flickr Zeng et al. [2022] | 16 | 86 | XVLM | Swin | XVLM Swin B |
| flava_full Singh et al. [2022a] | 70 | 86 | Other | vit | FLAVA ViT B 32 |
| openclip_vitL14_400m Ilharco et al. [2021] | 400 | 307 | Pure Contrastive | vit | OpenCLIP ViT L 14 |
| openclip_vitL14_datacomp_xl Gadre et al. [2023b] | 12800 | 307 | Pure Contrastive | vit | DataComp ViT L 14 |
| openclip_vitL14_2b Ilharco et al. [2021] | 2000 | 307 | Pure Contrastive | vit | OpenCLIP ViT L 14 |
| clip_vitL14 Radford et al. [2021b] | 400 | 307 | Pure Contrastive | vit | CLIP ViT L 14 |
| xvlm_coco Zeng et al. [2022] | 16 | 86 | XVLM | Swin | XVLM Swin B |
| openclip_vitB32_400m Ilharco et al. [2021] | 400 | 86 | Pure Contrastive | vit | OpenCLIP ViT B 32 |
| openclip_vitB32_2b Ilharco et al. [2021] | 2000 | 86 | Pure Contrastive | vit | OpenCLIP ViT B 32 |
| openclip_vitG14_2b Ilharco et al. [2021] | 2000 | 1011 | Pure Contrastive | vit | OpenCLIP ViT g 14 |
| openclip_vitbigG14_2b Ilharco et al. [2021] | 2000 | 1843 | Pure Contrastive | vit | OpenCLIP ViT G 14 |
| openclip_vitB16_2b Ilharco et al. [2021] | 2000 | 86 | Pure Contrastive | vit | OpenCLIP ViT B 16 |
| openclip_vitB16_400m Ilharco et al. [2021] | 400 | 86 | Pure Contrastive | vit | OpenCLIP ViT B 16 |
| opencoca_vitL14_2b Yu et al. [2022a], Ilharco et al. [2021] | 2000 | 307 | Other | vit | OpenCOCA ViT L 14 |
| opencoca_vitB32_2b Yu et al. [2022a], Ilharco et al. [2021] | 2000 | 86 | Other | vit | OpenCOCA ViT B 32 |
| negclip_vitB32 Yuksekgonul et al. [2023] | 400 | 86 | Negative CLIP | vit | NegCLIP ViT B 32 |
| clip_vitB16 Radford et al. [2021b] | 400 | 86 | Pure Contrastive | vit | CLIP ViT B 16 |
| clip_resnet50 Radford et al. [2021b] | 400 | 38 | Pure Contrastive | conv | CLIP ResNet50 |
| openclip_resnet101_yfcc Ilharco et al. [2021] | 15 | 56 | Pure Contrastive | conv | OpenCLIP ResNet101 |
| openclip_resnet50_yfcc Ilharco et al. [2021] | 15 | 38 | Pure Contrastive | conv | OpenCLIP ResNet50 |
| openclip_resnet50_cc Ilharco et al. [2021] | 12 | 38 | Pure Contrastive | conv | OpenCLIP ResNet50 |
| clip_resnet101 Radford et al. [2021b] | 400 | 56 | Pure Contrastive | conv | CLIP ResNet101 |
| clip_resnet50x4 Radford et al. [2021b] | 400 | 87 | Pure Contrastive | conv | CLIP ResNet50x4 |
| clip_resnet50x16 Radford et al. [2021b] | 400 | 167 | Pure Contrastive | conv | CLIP ResNet50x16 |
| clip_resnet50x64 Radford et al. [2021b] | 400 | 420 | Pure Contrastive | conv | CLIP ResNet50x64 |
| clip_vitB32 Radford et al. [2021b] | 400 | 86 | Pure Contrastive | vit | CLIP ViT B 32 |

Table 6: List of all the models used in evaluations with their corresponding dataset size, model size (number of parameters), learning objective, and architecture.

| Benchmark | Measure | Benchmark Type | Capability | Curated | Object Centric | Number of Classes |
|---|---|---|---|---|---|---|
| caltech101 [Fei-Fei et al., 2004] | zero-shot | object recognition | standard object recognition | False | True | 102 |
| cars [Krause et al., 2013] | zero-shot | object recognition | standard object recognition | False | True | 196 |
| cifar10 [Krizhevsky et al., 2009] | zero-shot | object recognition | standard object recognition | False | True | 10 |
| cifar100 [Krizhevsky et al., 2009] | zero-shot | object recognition | standard object recognition | False | True | 100 |
| clevr count [Johnson et al., 2017] | zero-shot | reasoning | counting | True | False | 8 |
| clevr distance [Johnson et al., 2017] | zero-shot | reasoning | spatial understanding | True | False | 6 |
| coco order [Yuksekgonul et al., 2023] | relation | relation | relations | False | False | 5 |
| countbench [Paiss et al., 2023] | zero-shot | reasoning | counting | False | False | 10 |
| country211 [Radford et al., 2021a] | zero-shot | object recognition | geographic diversity | False | False | 211 |
| cub [Wah et al., 2011] | zero-shot | object recognition | specifies classification | False | False | 200 |
| dmlab [Zhai et al., 2019] | zero-shot | reasoning | depth estimation | True | False | 6 |
| dollar street [Gaviria Rojas et al., 2022] | zero-shot | object recognition | geographic diversity | False | True | 60 |
| dspr orientation [Matthey et al., 2017] | zero-shot | reasoning | pose detection | True | False | 40 |
| dspr x position [Matthey et al., 2017] | zero-shot | reasoning | spatial understanding | True | False | 32 |
| dspr y position [Matthey et al., 2017] | zero-shot | reasoning | spatial understanding | True | False | 32 |
| dtd [Cimpoi et al., 2014] | zero-shot | texture | texture detection | True | False | 47 |
| eurosat [Helber et al., 2019, 2018] | zero-shot | non-natural images | satellite | False | False | 10 |
| fashion mnist [Xiao et al., 2017] | zero-shot | object recognition | character recognition | True | True | 10 |
| fgvc aircraft [Maji et al., 2013] | zero-shot | object recognition | standard object recognition | False | True | 100 |
| flickr30k order [Yuksekgonul et al., 2023] | relation | relation | relations | False | False | 5 |
| flowers102 [Nilsback and Zisserman, 2008] | zero-shot | object recognition | specifies classification | False | True | 102 |
| food101 [Bossard et al., 2014] | zero-shot | object recognition | standard object recognition | False | True | 101 |
| gtsrb [Stallkamp et al., 2012] | zero-shot | object recognition | standard object recognition | False | True | 43 |
| imagenet1k [Deng et al., 2009] | zero-shot | object recognition | imagenet | False | True | 1000 |
| imagenet9[Xiao et al., 2020] | zero-shot | robustness | natural transformations | True | True | 1000 |
| imagenet sketch [Wang et al., 2019] | zero-shot | non-natural images | rendition | True | True | 1000 |
| imageneta [Hendrycks et al., 2021b] | zero-shot | robustness | challenging imagenet | True | True | 200 |
| imagenetc [Hendrycks and Dietterich, 2019] | zero-shot | corruption | corruption | True | True | 1000 |
| imagenete [Li et al., 2023c] | zero-shot | robustness | natural transformations | True | True | 1000 |
| imageneto [Hendrycks et al., 2021b] | zero-shot | robustness | challenging imagenet | True | True | 200 |
| imagenetr [Hendrycks et al., 2021a] | zero-shot | non-natural images | rendition | True | True | 200 |
| imagenetv2 [Recht et al., 2019] | zero-shot | robustness | challenging imagenet | True | True | 1000 |
| inaturalist [Van Horn et al., 2018] | zero-shot | object recognition | specifies classification | False | True | 5089 |
| kitti distance [Geiger et al., 2012] | zero-shot | reasoning | depth estimation | False | False | 4 |
| mnist[LeCun et al., 1998] | zero-shot | object recognition | character recognition | True | True | 10 |
| objectnet [Barbu et al., 2019] | zero-shot | robustness | natural transformations | False | True | 113 |
| pcam [Veeling et al., 2018] | zero-shot | non-natural images | medical | True | False | 2 |
| pets [Parkhi et al., 2012] | zero-shot | object recognition | specifies classification | False | True | 37 |
| places365 [Zhou et al., 2017] | zero-shot | object recognition | scene recognition | False | False | 365 |
| pug imagenet [Bordes et al., 2023] | zero-shot | object recognition | standard object recognition | False | True | 151 |
| renderedsst2 [Radford et al., 2021a] | zero-shot | object recognition | character recognition | True | True | 2 |
| resisc45[Cheng et al., 2017] | zero-shot | non-natural images | satellite | False | False | 45 |
| retinopathy [Wang and Yang, 2018] | zero-shot | non-natural images | medical | False | False | 5 |
| smallnorb azimuth [LeCun et al., 2004] | zero-shot | reasoning | pose detection | True | False | 18 |
| smallnorb elevation [LeCun et al., 2004] | zero-shot | reasoning | spatial understanding | True | False | 9 |
| stl10 [Coates et al., 2011] | zero-shot | object recognition | standard object recognition | False | True | 10 |
| sugarcrepe [Hsieh et al., 2024] | relation | relation | relations | False | False | 2 |
| sun397 [Xiao et al., 2010] | zero-shot | object recognition | scene recognition | False | False | 397 |
| svhn [Netzer et al., 2011] | zero-shot | object recognition | character recognition | False | True | 10 |
| vg attribution [Yuksekgonul et al., 2023] | relation | relation | relations | False | False | 2 |
| vg relation [Yuksekgonul et al., 2023] | relation | relation | relations | False | False | 2 |
| voc2007 [Everingham et al.] | zero-shot | object recognition | standard object recognition | False | True | 20 |
| winoground [Thrush et al., 2022a] | relation | relation | relations | False | False | 2 |

Table 7: List of all the benchmarks used in evaluations with their corresponding dataset type, capability, number of classes, whether they are curated and whether they are curated object centric.

