# OpenReview forum: "UniBench: Visual Reasoning Requires Rethinking Vision-Language Beyond Scaling"
_NeurIPS.cc/2024/Datasets_and_Benchmarks_Track — NeurIPS 2024 Track Datasets and Benchmarks Poster_

### Official Review · Reviewer_1p61 · 2024-06-26
**Review for Submission2130**

**Rating:** 7
**Confidence:** 4
**Correctness:** Lack of details of evaluation methods…

**Review:**

Strengths

1. The paper evaluates a wide range of models and datasets.
2. The evaluation process is time-efficient.
3. Comprehensive evaluation of VLMs is crucial for understanding their capabilities and limitations.

Weaknesses

1. Lack of related-works section.

The paper lacks a comparison to previous evaluation works on VLMs. The paper didn't explain why building such a framework is challenging, as calculating the similarity between the image representation and text label is simple.

2. Lack of detailed designation of the framework.

As the main contribution is the systematic framework, the paper didn't show any detailed information on the framework's design. It seems that the framework merely contains two parts: models and datasets. It is simple and straightforward to evaluate a VLM.

3. Taxonomy of Benchmarks

Previous works like MMBench, MMMU, and SEEDBench offer comprehensive zero-shot classification evaluation datasets for VLMs. Those works' evaluated domains and dimensions are broader and more thorough than those presented in this paper.

**Strengths:**

Please see the Review.

**Additional Feedback:**

N/A

**Clarity:**

Lack of details of the framework designation. The supplementary is not provided.

**Documentation:**

N/A

**Limitations:**

No potential negative societal impact.

**Opportunities For Improvement:**

Please see the Weaknesses.

**Relation To Prior Work:**

It is not clear since there is no discussion on related works.

**Summary And Contributions:**

The paper introduces UniBench, an evaluation framework for the unified evaluation of vision-language models, i.e., CLIP, and EVA. The authors evaluate 59 VLMs across 53 benchmarks, comparing various models. They conclude that increasing model size or training data does not significantly improve performance in visual relations and reasoning tasks.

---

> ### Author Rebuttal · Authors · 2024-08-16
>
> Thank you for your thorough review and valuable feedback on our submission. We appreciate the time you invested in evaluating our work and your insightful comments, which have helped us identify areas for improvement.
>
> # Lack of Related Works and Taxonomy of Benchmarks
> While we do include a brief discussion of related work in Section 1 and 3 in our initial submission, we appreciate the reviewer’s feedback that a dedicated, expanded related works section would better help contextualize our contribution. To address this, we now include a dedicated related works section with a comprehensive comparison to existing VLM benchmarks. We now clearly delineate how UniBench differs from these works in terms of scope, and the variety of tasks covered.  In summary:
>
> While existing benchmarks such as MMBench, MMMU, and SEEDBench provide comprehensive zero-shot classification evaluation datasets for VLMs, they are primarily geared towards LLM-style VLMs (e.g., Llava) that generate tokens or text as output. Although these benchmarks can effectively evaluate LLM-style VLMs, they are not suitable for evaluating CLIP-like VLMs, which focus on vision-language classification and understanding capabilities. The ability to evaluate CLIP-like VLMs is critical to assess the progress of the field over the past several years since the release of CLIP.
>
> Moreover, benchmarks like MMMU primarily assess specialized knowledge from academic disciplines such as engineering, art, and history. In contrast, our work with UniBench aims to benchmark generic vision-language classification and understanding capabilities, providing a more comprehensive vision evaluation framework for VLMs. By focusing on traditional zero-shot tasks and predefined choices, UniBench enables an apples-to-apples comparison of progress over the past few years, shedding light on promising directions for future research.
>
> Finally, it's worth noting that UniBench has a significantly higher scale in the number of evaluations, containing 5.9 Million samples, compared to existing benchmarks such as SEEDBEnch (20k samples), MMMU (11.5k samples), and MMBench (3k samples). While also being comparable in the number of capabilities being evaluated, containing 17 capabilities, compared to existing benchmarks such as SEEDBEnch (12), MMMU (6), and MMBench (20). In short, **UniBench contains more than 100x the number of samples** while having a comparable number of taxonomies as these existing benchmarks. This larger scale allows for more robust and reliable evaluations of VLMs.
>
> In addition to the above mentioned works, we incorporate further comparisons with existing works. As well as the unique insights we provide into the limits of scaling and promise of tailored learning objectives not surfaced in other work.
> # Lack of Details of the Framework Designation
>
> We understand that the initial submission lacked detailed information on the framework's design. We have revised the code documentations to include a detailed description of the architecture and components of UniBench.
>
> * “The supplementary is not provided”: we did include a supplementary codebase and appendix with additional figures/analysis in our initial submission that we’d like to draw your attention to. Based on reviewers’ feedback, we’ve also included additional figures/analysis in an additional rebuttal PDF. Finally, we plan to also open-source the entire framework.
> * Complexity of evaluation: We would like to highlight that it is challenging to implement a consistent evaluation framework, across a large number of benchmarks (at a scale of more than 100x prior work). This is also evidenced by prior efforts such as MMMU at unified benchmarks for specialized knowledge as well as the disparate evaluations found in VLM papers. It is challenging because new VLM papers provide piecemeal benchmarks in their analyses. In addition to the library contribution, we also offer insights into the limits of scale and promise of tailored learning objectives/data quality that we'd like to point out other reviewers found extremely valuable—we believe beyond the ease of use of our library, these insights could steer future advances in vision-language modeling.
> * UniBench design summary: UniBench is designed to be extensible and flexible, allowing users to easily add new benchmarks and models. The framework consists of multiple components, including data loaders, model interfaces, results caching and evaluation metrics, which work together to provide a comprehensive evaluation suite for VLMs. Furthermore, we believe that the simplicity and straightforwardness of our framework are actually strengths, rather than weaknesses. By providing an easy-to-use interface, we enable researchers to focus on developing their models, rather than spending time on implementing evaluation protocols and comparing with a large set of existing models. In addition, we would like to highlight that our framework has been designed with scalability in mind. We have implemented 59 VLMs across 53 benchmarks to compare across, and our framework has demonstrated its ability to handle large-scale evaluations. Lastly, the framework allows for a distilled set of representative benchmarks that can run in less than 5 minutes on a single A100 GPU, allowing for insightful evaluation of VLMs during training.
> # Evaluation Methods
> To address the concern regarding the lack of details of evaluation methods, we have provided a detailed description of the architecture and components of UniBench in our revised code documentation.
> Regarding the specific evaluation methods used, we employ a standard zero-shot classification protocol, where the model is presented with a test image and a set of text labels/captions, and it must predict the correct label/caption. We also provide an analysis of the results, including correlations between benchmark performances (Figure 12) and performance of 59 VLMs across 53 benchmarks to compare across (Figure 2).

---

> > ### Comment · Reviewer_1p61 · 2024-08-26
> >
> > Thank you to the authors for their response.
> >
> > After reviewing the rebuttal and considering the comments from other reviewers, I believe the contribution of this paper is sufficient. However, the paper would benefit from a more extensive discussion on the evaluation of VLMs and a clearer positioning of the paper's contribution. Overall, I am raising my score to 7.

---

### Official Review · Reviewer_qzJ1 · 2024-07-17
**Solid work but limited scope**

**Rating:** 7
**Confidence:** 3
**Clarity:** The paper is written adequately, and …

**Review:**

Standardizing benchmarks for VLMs is an important and timely contribution.
The authors did a reasonable job of collecting and categorizing datasets across various tasks and domains.
Similarly, the set of evaluated VLMs (59) is comprehensive.
However, due to my lack of deeper experience with the VLM landscape, I also cannot judge whether there are any major omissions.
Making both datasets and models all accessible with a common interface should help research progress faster (of course bearing the danger of https://xkcd.com/927/).
I like that the authors defined a representative subset of tasks that can be evaluated faster than the full set.

The analysis of the results is adequate.
I appreciated the correlations between datasets in Figure 12.
In addition, I would have wished for an experiment for scaling both data and model size at the same time, as significant scaling effects are usually observed when increasing both together.
Moreover, a grouping of the results by different model properties would be interesting, for example training objective, internal model resolution (i.e. patch size), or diversity of the training datasets (i.e., to gauge possible correlations between properties of the training data and downstream performance).
Possibly I missed it, but I could not find model performance on the limited task set, and how it correlates with the results on the full task set.

The biggest limitation of the work seems to be the restriction to contrasting single vector representations as a way to derive the output.
Even though this corresponds to standard protocol when evaluating CLIP-like models, it also restricts the possible performance achievable on the tasks the authors evaluate.
In particular, it is somewhat unsurprising that a single vector representation does not lend itself well to reasoning tasks where relations between objects need to be considered.
More flexible interfaces, e.g. where input is not just a single vision vector representation, and the output can be specified in natural language, would allow to take more (and future) models into account that have better means to solve tasks where CLIP-like models struggle.

Another limitation could be the lack of a true open-world dataset for object recognition; as far as I can see, all the datasets in that category have a set of predefined classes. Even though collectively, they of course constitute a large set of classes, this does not fully capture the challenges of open-world recognition.
Related, a further interesting benchmark type could be long-tail problems with rare objects, which is a problem of strong practical relevance, and thus judging the performance of VLMs on that is important.

Overall, I find the work to be solid in its core contribution, but somewhat limited with regards to its future potential; in that sense, the work is a nice consolidation of existing work, but not particularly forward looking.

**Strengths:**

- The UniBench benchmark is a nice addition to the field of VLMs, providing a standardized and comprehensive evaluation suite (however, possibly too limited in scope)
- The authors gathered a broad set of tasks and domains (however, possibly missing true open world and long-tail problems)
- An analysis of current CLIP-style VLM capabilities and limitations. While the results and conclusions have appeared scattered in different places already, it's good to reproduce them once more and gather them together.
- The fine-grained model capabilities enable a more nuanced understanding of model performance, possibly leading to more targeted improvements to VLMs

**Additional Feedback:**

Instead of computing an average over datasets per category, could the authors consider taking a weighted average normalizing by task difficulty (e.g. using the performance of some baseline model)? I imagine the datasets in each category to vary in terms of their difficulty, and thus the average performance could be distorted towards the easy tasks.

**Correctness:**

Evaluation protocol seems to follow the standard in the literature.
The codebase looks reasonably clean.
I assume that data and model download is handled automatically by the library, it would be great if the authors could explicitly clarify.
I would have wished for a more detailed description in the paper about the exact interface needed for extending the benchmark with a new dataset (Code Snippet 1 suggests any standard torchvision datasets works, but surely there is more to be provided, e.g. a set of prompt templates).

**Documentation:**

Code and documentation is provided.

**Limitations:**

The limitations section is very sparse, although the authors admit to the problem of a single evaluation protocol.
A wider discussion of possible extensions would be appreciated, which could also change the picture with regards to the conclusion the authors draw from their benchmark results.

**Opportunities For Improvement:**

- The model interface is limited to a single vector representation on the vision and language side. This severely restricts the type of models that can be evaluated to the currently popular CLIP-style paradigm.
- The paper would benefit from experiments that simultaneously scale both data and model size systematically. This could show scaling effects that are not evident when scaling only one dimension.
- Grouping results by different model properties, such as training objective, internal model resolution, or diversity of training datasets, could provide deeper insights into how these factors influence performance.
- Possible extensions to the set of datasets and tasks to make the benchmark even more comprehensive (open-world recognition and long-tail problems)
- Broader discussion of limitations and related work

**Relation To Prior Work:**

The paper fully misses a discussion of related work. This makes it hard to judge how original its contributions are and how they fit into the wider research landscape, both in terms of dataset curation and model evaluation. For example, the limitations of with VLMs with regard to reasoning, relations or character recognition have been noted before (e.g., [1]); the paper would benefit from a discussion of those.

[1] Tong et al.: Eyes Wide Shut? Exploring the Visual Shortcomings of Multimodal LLMs. CVPR 2024.

**Summary And Contributions:**

This paper contributes the UniBench benchmark, a collection of 53 previously existing datasets for vision-language understanding.
The datasets cover different tasks (object recognition, spatial reasoning) and robustness to various distribution shifts (corruptions, non-natural images).
The authors also define 17 model capabilities needed for the individual datasets to aid more nuanced understanding of model performance.
In the paper, 59 VLMs are then evaluated on this benchmark, including ablations to data and model scale.
The findings indicate that current VLMs struggle in particular on reasoning tasks and character recognition.
The authors release code that covers download, execution and evaluation for all datasets and models.

---

> ### Author Rebuttal · Authors · 2024-08-16
>
> Thank you for your thoughtful suggestions. We’re glad you agree that standardizing benchmarks is an important problem, found our benchmark practical/comprehensive, and found our insights into model capabilities useful.
>
> # Support for Natural Language Outputs
> Based on your feedback, we’ve extended UniBench to support models with natural language outputs (following [1]). We include new evaluations for Llava:
> | model name|corruption|non-natural images|object recognition|reasoning|relation|robustness|texture|
> |:-|-:|-:|-:|-:|-:|-:|-:|
> |llava_1_5_13b|0.31|0.50|0.36|0.11|0.41|0.24|0.34|
> |llava_1_5_7b|0.29|0.51|0.32|0.12|0.42|0.23|0.28|
>
> [1] Matsuura, M., Jung, Y. K., & Lim, S. N. (2023). Visual-LLM Zero-Shot Classification.
>
> # Scaling Data and Model Size Systematically and other factors influencing performance
> We agree simultaneously scaling data and model size is insightful as practitioners often scale both. Based on your feedback, we’ve conducted a new analysis of simultaneous scaling by examining progress on ImageNet (Figure 1 of the rebuttal PDF). Our findings indicate even with simultaneous scaling, VLMs still struggle on reasoning and relational understanding tasks. These results are inline with Table 1 in our initial submission.
> We also explored the impact of other model properties on performance such as the effect of object-centric vs non-object centric benchmarks, model resolution, and model architecture (conv vs ViT). However, we found that these factors did not have a significant impact on performance. Although we did not include these analyses in our initial submission, we plan to add analyses of these findings to Section A.6 of our camera-ready draft.
> # Extending the set of benchmarks in UniBench
> We would like to highlight that our current benchmark set already includes many benchmarks that are not class-based. For example, benchmarks assessing visual relations, reasoning, and pose estimation probe models’ understanding of relationships between objects, a key aspect of open-world recognition. We also include benchmarks such as iNat-2017 (5k animals), which encompass rare and diverse classes useful for studying VLMs’ capabilities on long-tail problems. Lastly, we updated the documentation to provide an extended code snippet example of how to add new benchmarks to UniBench (see below).
> # Lack of Related Works
> We appreciate the reviewer’s feedback and now provide a dedicated, expanded related works section would better contextualize our contribution, which we summarize below:
>
> While existing benchmarks such as MMBench, MMMU, and SEEDBench provide comprehensive zero-shot classification evaluation datasets for VLMs, they are not suitable for evaluating CLIP-like VLMs, which is critical to assess the progress of the field over the past several years since the release of CLIP.
>
> Moreover, our work with UniBench aims to benchmark generic vision-language classification and understanding capabilities, providing a more comprehensive vision evaluation framework for VLMs, while many existing benchmarks such as MMMU focus on specialized knowledge of academic disciplines.
>
> Finally, UniBench has a significantly higher number of evaluations, containing 5.9 Million samples, compared to the existing benchmarks such as SEEDBEnch (20k samples), MMMU (11.5k samples), and MMBench (3k samples), while also being comparable in the number of capabilities (17). In short, **UniBench contains more than 100x the number of samples** while having a comparable number of taxonomies as these existing benchmarks. This larger scale allows for more robust and reliable evaluations of VLMs.
> # Other Suggestions
> ## Weighted Average Performance
> Based on your recommendation, we conducted a new analysis, computing the weighted average performance for each model using CLIP B/32 as the baseline model performance (as a proxy for task difficulty; included in Rebuttal, Figure 3). Our results show normalizing by task difficulty does lower the overall performance, but not alter model rankings. We will include this analysis in the revised version of our paper in Section A.7.
> ## Implementation Details
> We would like to clarify that the UniBench library does indeed handle data and model download automatically (via HuggingFace), making it easy for users to evaluate their models on a wide range of benchmarks without having to worry about the logistics of data and model management. Furthermore, all model responses of the 59 VLMs in the paper are cached and downloaded by the library automatically, making reproducibility and expanding on the current work easy.
> ## Extended Code Snippet
> We have updated the documentation with an extended code snippet of how to add new benchmarks to UniBench. Below is the updated code snippet for adding a torchvision dataset:
> ```
> from unibench import Evaluator
> from unibench.benchmarks_zoo import ZeroShotBenchmarkHandler
> class_names = ["T-shirt/top", …]
> templates = ["an image of {}"]
> handler = partial(ZeroShotBenchmarkHandler, classes=class_names, templates=templates, )
> eval = Evaluator()
> eval.add_benchmark(FashionMNIST, handler )
> ```
> ## Limited task set correlation with full task set
> We intentionally selected the fast subset of UniBench by identifying benchmarks with the highest correlations for each benchmark type, which implies by definition the benchmark we selected for each type is most correlated. We show the correlations for each selected benchmark in Table 1 and also show full correlations in Figure 12.
> ## Forward Looking Insights
> * Support for natural language output: We have expanded UniBench to include more recent LLM-style models such as Llava.
> * Insights into the limits of naive scaling and promise of tailored learning objectives (Section 4.3)
> * Practical guidance for practitioners on which models to select: UniBench can help suggest VLMs to practitioners on which models to select and what attributes in models that can play a crucial role in improving VLM performance on specific tasks in table 1.

---

> > ### Comment · Reviewer_qzJ1 · 2024-08-26
> > **Thank you**
> >
> > I thank the authors for their response and their additional experiments.
> >
> > The authors alleviated my main concern, the rigidity of the model interface, only partially. The new inclusion of Llava style models is great and complements the paper well; however, the evaluation is ultimately still based on comparing single vector representations (in this case, between two text embeddings). On the vision side, the input representation is similarly still restricted to a single vector. This limits the kind of models and tasks that can be evaluated and the performance that can be achieved on them. Thus, the comment of my rebuttal of this benchmark not being particularly forward-looking, still stands.
> >
> > However, with all the new additions, I think this is solid work that should be accepted at the conference. I raise my score to 7.

---

### Official Review · Reviewer_GrCg · 2024-07-23
**A strong practical contribution with many insights on scaling trends**

**Rating:** 7
**Confidence:** 4

**Review:**

UniBench is a strong contribution that keeps in mind the research practitioners and what drives evaluation progress in reality: unified setups, a simple few lines of code and quick runtimes.
On top, there are several insights and recommendations for researchers, informing the design of training data and learning pipelines.

**Strengths:**

It is good the authors thought of making a distilled, representative set of benchmarks.
A lot of nice insights and arguments for why this is needed from a day-to-day practical standpoint of a researcher, and the messy reality of how recent VL papers showcase their results.

Showing code snippets on how to use UniBench is unusual but very fitting and demonstrates the main contribution: ease of use for other researchers!

There are some nice in-depth studies, especially the case study 3.2 seems well conducted and tests for a lot of possible explanations. Overall, the authors make a compelling case that naive scaling is not enough for several capabilities.

The correlation-between-benchmarks insights are very useful for practitioner and can inform us whether relying on a few semi-randomly chosen benchmarks is a good strategy, for example: “While ImageNet
maybe a tempting candidate as it correlates with many benchmarks, for many others, specifically 18
of the 53 benchmarks, ImageNet performance is poorly or negatively correlated Appendix Figure 11.” Moreover, this means the distilled smaller version of UniBench is empirically justified with correlation and not just by intuition.

**Additional Feedback:**

One small question: where is the collection of benchmarks stored? I assume since the user only has to load the library, they don’t have to explicitly download any data. Is it based on huggingface datasets or works in a similar manner?

**Clarity:**

Overall, the practical motivation is very convincingly formulated. The whole paper is easy to follow and does not try to confuse the reader with overly strong claims.

**Correctness:**

The paper is thorough and reports a lot of details and correlations. Due to the large-scale nature, the results are hopefully quite robust.

**Documentation:**

For 3.1 (analysis section), it would be good to include which model(s) exactly you compared for data/model sizes.

The repository is well documented and easy to follow, and the ease of use for the library is demonstrated.

**Ethics:**

No, it is ethically sound.

**Limitations:**

The limitations are discussed in a short but sufficient manner.

**Opportunities For Improvement:**

It seems that UniBench is somewhat skewed towards axes of traditional vision tasks. This should be disclosed upfront, that UniBench is not all encompassing for all VL tasks since it is framed as evaluating Vision Language capabilities in the main paper (I see that the title mentions visual reasoning). The contribution is still strong but this makes it easier for people to know what exactly they get. The only benchmarks that are true vision+language understanding are relation and some of reasoning it seems.

**Relation To Prior Work:**

This part is missing and seems important: I am sure there must have been similar efforts of unifying the chaos in VL? Are they out-dated? What are they missing? Including this in the camera ready would make the paper stronger and comprehensive.

**Summary And Contributions:**

The paper unifies a very large number of 53 vision-and-language benchmarks into UniBench, and evaluates 59 VL models on it. By making it easy for future researchers to use, it alleviates the burden of setting up different protocols and interpreting the results, e.g. regarding scaling trends.
Due to the large-scale nature of the study, the authors can confidently draw conclusions on certain trends and failure modes, i.e. that scaling does not help much for tasks such as relational reasoning or digit recognition, or that data quality is a key factor over quantity.

---

> ### Author Rebuttal · Authors · 2024-08-16
>
> Thank you for your thoughtful suggestions. We absolutely agree that strong drivers of research progress in practice are as you mention “unified setups, a simple few lines of code and quick runtimes.” We wholeheartedly agree and set out to design UniBench with these values in mind.
>
> # UniBench is skewed towards traditional vision tasks
> Thank you for pointing out the skewness of UniBench towards traditional vision tasks. We agree that the contributions of UniBench could benefit from better defining the emphasis of the benchmarks we selected. We do focus on zero-shot tasks that don’t rely on a text-decoder with chat-capabilities (a feature more common in recent model with pretrained text decoder backbones) for two reasons: 1) to ensure we can measure progress in apples-to-apples fashion going back several years to CLIP models to understand which aspects of training have been most crucial to advances in model capabilities 2) because evaluation is more objective with predefined choices that do not require parsing open-dialogue response. While this of course does skew the set of tasks we cover, across the 50+ benchmarks we evaluate we cover not only standard vision recognition tasks, but also more nuanced grounding benchmarks that include relations across objects (both spatial and in terms of objective interactions), attributes, and text compositionality. Of course, UniBench is not limited to some benchmarks or models but users can add further benchmarks. Nonetheless, we agree we should’ve framed UniBench more clearly in light of our choice of benchmarks. We will revise the framing in our abstract, introduction, conclusions, and highlight that further in our experimental section to appropriately account for this choice in our benchmarks. Thank you for this suggestion
> # Lack of Related Works
> We appreciate the reviewer’s feedback that a dedicated, expanded related works section would better help contextualize our contribution. To address this, we now include a dedicated related works section with a comprehensive comparison to existing VLM benchmarks. We now clearly delineate how UniBench differs from these works in terms of scope, and the variety of tasks covered. We summarize some key differences to existing works below:
>
> While existing benchmarks such as MMBench, MMMU, and SEEDBench are also evaluation datasets for VLMs, they are primarily geared towards LLM-style VLMs (e.g., Llava) that generate tokens or text as output. Although these benchmarks can effectively evaluate LLM-style VLMs, they are not suitable for evaluating CLIP-like VLMs, which focus on vision-language classification and understanding capabilities. The ability to evaluate CLIP-like VLMs is critical to assess the progress of the field over the past several years since the release of CLIP.
>
> Moreover, benchmarks like MMMU primarily assess specialized knowledge from academic disciplines such as engineering, art, and history. In contrast, our work with UniBench aims to benchmark generic vision-language classification and understanding capabilities, providing a more comprehensive vision evaluation framework for VLMs. By focusing on traditional zero-shot tasks and predefined choices, UniBench enables an apples-to-apples comparison of progress over the past few years, shedding light on promising directions for future research.
>
> Finally, it's worth noting that UniBench has a significantly higher scale in the number of evaluations, containing 5.9 Million samples, compared to existing benchmarks such as SEEDBEnch (20k samples), MMMU (11.5k samples), and MMBench (3k samples). While also being comparable in the number of capabilities being evaluated, containing 17 capabilities, compared to existing benchmarks such as SEEDBEnch (12 capabilities), MMMU (6 capabilities), and MMBench (20 capabilities). In short, UniBench contains more than 100x the number of samples while having a comparable number of taxonomies as these existing benchmarks. This larger scale allows for more robust and reliable evaluations of VLMs.
>
> In addition to the above mentioned works, we incorporate further comparisons with existing works. As well as the unique insights we provide into the limits of scaling and promise of tailored learning objectives not surfaced in other work.
> # Including models compared in section 3.1
> Thank you for pointing this out. For investigating the impact of training dataset size (Figure 3, left panel), we used OpenCLIP models trained on LAION 2B, DataComp S, DataComp M, DataComp XL, LAION 400M, and MetaCLIP 2.5B, while maintaining a fixed architecture, learning paradigm, and model size (ViT-B/32). For investigating the impact of model size (Figure 3, right panel), we used OpenCLIP models trained on LAION 400M with ViT-B/32, ViT-L/14, and ViT-G/14 architectures, while maintaining a fixed architecture, learning paradigm, and dataset size. We will make sure to include this information in the caption under figure 4 and 5 and update the description in section 3.
> # Does UniBench require users to download benchmarks or is it based on HuggingFace?
> This is a great point we will make sure to highlight in the camera-ready draft. The collection of benchmarks and models in UniBench are based on HuggingFace, users do not need to individually download or preprocess each benchmark. Furthermore, all model responses of the 59 VLMs across 50 benchmarks in the paper are cached and downloaded by the library automatically and can be loaded in seconds, making reproducibility and expanding on the current work easy. UniBench handles all complexities of downloading and parsing the benchmarks without the need for additional tools.

---

> > ### Comment · Reviewer_GrCg · 2024-08-19
> >
> > Thank you for the thorough rebuttal and addressing a lot of concerns and questions!
> >
> > Two things have made me reconsider my score and raise it to 7:
> >
> > 1. The additions of related work, support for VLM outputs, ... are convincing
> > 2. Also, re-emphasizing the need to study scaling and trends over several years since CLIP came out in a unified way, was helpful to be reminded of

---

### Official Review · Reviewer_eN9n · 2024-07-24
**Review of  UniBench: Visual Reasoning Requires Rethinking Vision-Language Beyond Scaling**

**Rating:** 9
**Confidence:** 4
**Correctness:** No concerns with the approach/method.

**Review:**

Benchmarking VLMs is a very active area of research and there are a large number of benchmarks covering a variety of tasks. In this context, it becomes burdensome for research scientists to meaningfully benchmark a new model as one would have to interface several different benchmarks and also coagulate the observations in a useful manner. Authors have effectively addressed this problem and the library they have developed makes the benchmarking exercise a simpler one. This is a substantially useful contribution to the field.

**Strengths:**

This paper has several strengths:
1. The library developed significantly reduces the effort to benchmark a new VLM model.
2. Authors have demonstrated several key limitations of the existing vision language models, ex. chance level performance for well solved  tasks such as MNIST
3. Authors have demonstrated that reasoning and relationship comprehension were not improving with increased dataset size or model size, may be the bitter lesson does not hold in this context.

**Additional Feedback:**

I would like to thank the authors for their efforts. The observed stagnation in terms of reasoning performance despite the data scaling is very interesting.

**Clarity:**

The paper is very well written, there might one or two types, please run a spell checker.

**Documentation:**

yes

**Ethics:**

No concerns.

**Limitations:**

This is a very well written paper, it would be great if authors could further explore the causal factors associated with the trends observed in the paper.

**Opportunities For Improvement:**

It would be great if authors could further explore some of the issues:
1. What is the causing the models to exhibit such a variability in the performance for simple MNIST task shown in Figure 5?
2. Why is the performance degrading in tasks such as depth estimation even as the recognition improves when datasets are scaled?

**Relation To Prior Work:**

The literature is adequately discussed.

**Summary And Contributions:**

In this paper authors have developed a library to systematically compare and contrast vision language models on 53 benchmarks. They have evaluated 59 VLMS using this library and demonstrated several key trends  in the performance of the models. Authors have also distilled the benchmarks and identified a representative subset that significantly reduced the evaluation time.

---

> ### Author Rebuttal · Authors · 2024-08-16
>
> We thank the review for their thorough review and valuable feedback on our submission. We appreciate the time and effort invested in evaluating our work.
> # Variability in Performance on MNIST
>
> We thank the reviewer for pointing out the variability in performance on MNIST shown in Figure 5. To address this, we have conducted a new analysis investigating the effect of scaling both data and model size on MNIST performance (Figure 2 of the rebuttal PDF).
>
> We found that while scaling data size improves performance on MNIST, scaling model size does not have a significant impact on performance. This suggests that the variability in performance on MNIST could be largely due to the differences in data size used during training, rather than model size. We plan to include this additional plot along with description to explain this variability in Section A.8 of the camera-ready draft.
> # Why does performance on depth estimation degrade while recognition improves?
>
> We appreciate the reviewer’s careful note of this trend. We found depth estimation, along with other finer-grained tasks such as spatial understanding and pose detection, do not improve with scale relative to standard object recognition and other coarse-grained tasks. While this is an open research question, our intuition is that signal for coarse-grained tasks such as classification is more directly available in the image-caption pairs used for training than that needed for finer-grained discernment of tasks such as spatial awareness, which is why we see improvements with data scale on these coarse grained tasks. To further probe this question, we isolated the effect of both model and data scale in Figure 2 of the rebuttal PDF. We found that while data scale in fact does not improve finer-grained tasks such as depth estimate, scaling the model did help slightly, which we conjecture is due to the larger model capacity being able to better extract depth signals from the available data.
>
> # Typos
>
> Finally, we’ll be sure to revise this paper with a spell-checker to catch any remaining typos. We appreciate the reviewer’s careful reading, these thoughtful suggestions, and took notice of the main message of our work regarding the stagnation of scale for capabilities such as relations and reasoning.

---

### Author Rebuttal · Authors · 2024-08-17

Thanks to all the reviewers for your time and effort during the review process. We appreciate that you found our work well written, insightful, and novel, and we’re glad that there is excitement about UniBench. Specifically, reviewers found
* “UniBench is a strong contribution that keeps in mind the research practitioners and what drives evaluation progress in reality: unified setups, a simple few lines of code and quick runtimes. “—eN9n24
* “Comprehensive evaluation of VLMs is crucial for understanding their capabilities and limitations.”—1p6125
* “The paper is thorough and reports a lot of details and correlations. Due to the large-scale nature, the results are hopefully quite robust.”—GrCg23

We have addressed each reviewer's comments separately, submitted a rebuttal document, and updated our code/documentations. If you find our responses satisfactory, we kindly request that you reconsider your score. Additionally, if reviewers have any further questions or concerns, we are more than happy to discuss these concerns further.

Based on reviewers’ feedback, we’ve made a considerable effort to improve the quality of our work:
* Included a dedicated ‘Related Works’ section with explicit comparisons to existing VLM benchmark suites:
  * Comprehensive set of zeroshot vision oriented benchmarks as opposed to specialized domain knowledge (MMMU) or model-curated (MMVP) datasets
  * Larger scale evaluation with 100x more samples than many existing benchmarks in a unified easy to use library
  * Ability to assess zero-shot discriminative models to track progress over the past several years since the release of CLIP
  * Scientific insights into drivers and limits of progress is vision-language modeling approaches
* Five additional experiments with new analysis using UniBench
  * New analysis isolating the effect of model and data scaling on digit recognition and depth estimation performance, offering practical insights about which levers are useful for which capability (with model size benefiting depth estimation; and data scale benefiting digit recognition) (new plots in additional PDF)
  * New analysis of simultaneous scaling effects of data and model size on performance of the 7 benchmark types
  * New results on Llava and support for VLMs with natural language outputs (see library improvements below)
  * Added results to exploring the impact of other model properties on performance such as the effect of object-centric vs non-object centric benchmarks, model resolution, and model architecture
  * We conducted a new analysis, computing the weighted average performance for each model using CLIP B/32 as the baseline model performance
* Library improvements
  * Additional support for evaluations of VLMs with language outputs (and new experiments with Llava)
  * Added HuggingFace accessibility of all evaluations, including model responses on benchmarks, that can be downloaded through UniBench or directly from HuggingFace.
  * Included a detailed description of the architecture and components of UniBench.
  * Added Expanded Code Snippet for adding a new benchmark
  * Added 20 new VLMs to UniBench source code, bringing total to of 79 models

Thanks to these improvements based on reviewers’ suggestions, we believe UniBench would be a valuable resource for the research community as a practical framework unifying the disparate approaches to benchmarking vision-language models along with insights into the limits of scaling and promise of tailored learning objectives to steer research into advancing the capabilities of vision-language models.

---

### Decision · Program_Chairs · 2024-09-26

**Decision:**

Accept (Poster)

**Comment:**

Reviewers unanimously recognized the work as a valuable contribution to vision-language modeling. The unified evaluation framework for benchmarking multiple VLMs across a wide range of tasks is useful to have. One reviewer highlighted the practical benefits of UniBench’s simple setup and quick runtime, emphasizing its utility for research practitioners. Another praised the comprehensive evaluation of VLM capabilities and limitations, noting its large-scale nature and robust results.

However, some reviewers pointed out limitations, including the need for a broader exploration of related works and the skewed focus towards traditional vision tasks. Despite these critiques, the paper was well-received for its insights into model performance trends, its ease of use, and its potential as a practical tool for advancing VLM research. Congratulations!